# A high-throughput conditioned-media-based screening system identifies inhibitors of aggregation induced by iPSC-secreted amyloid β

Masahiro Kuragano[1], Naoki Nishishita[1,2,5] ✉, Koki Araya [1], Akira Kobayashi[2], Taro Q. P. Noguchi[3], Kenichi Watanabe[4], Shinya Watanabe[1], Stefan Baar[1], Koji Uwai [1] & Kiyotaka Tokuraku [1,5] ✉

In early drug discovery, in vitro screening is frequently used, but selected candidates often fail in vivo. Induced pluripotent stem cell (iPSC)–based disease models offer improved physiological relevance; however, the high costs of media and differentiation procedures limit large-scale testing. Here, we develop a high-throughput conditioned-media-based screening system—the High-throughput screening technology for Aggregation Inhibitors of Diseased cell-derived Aggregative Proteins (HaiDap) system—to identify inhibitors of aggregation induced by iPSC-secreted amyloid β (Aβ). Using conditioned media derived from differentiated iPSCs of a male Alzheimer's disease patient, we screen extracts from 22 edible plants. Whereas PBS-based assays showed 40.9% (9/22) apparent selectivity, the HaiDap system demonstrates higher specificity (13.6%; 3/22). All three identified extracts (*O. aristatus*, *S. aromaticum*, and *G. yesoense*) significantly delay Aβ aggregation on neuronal surfaces in an iPSC-based assay. These findings suggest that the HaiDap system enables efficient, accurate, and low-cost screening of amyloid aggregation inhibitors.

In recent years, the cost of drug development has skyrocketed. In drug development, candidate substances are screened for disease treatments in vitro at the first stage. Then, during the second stage, promising candidate substances are selected, and their effectiveness is evaluated using an in vivo assay with disease model animals. Recently, disease modeling based on dish technology using induced pluripotent stem cells (iPSCs) and genome editing technology, like CRISPR, are useful tools for developing new drugs and treatment methods for various diseases. Furthermore, the application of iPSCs for screening candidate drugs during drug discovery is also expected to be a tool to eliminate extrapolation to experimental animals and humans, which

has been a challenging objective until now. Most screening systems are performed in vitro at an early stage of drug discovery, but a problematic aspect is that candidate substances selected by the in vitro test do not always display an effect in cell-based assays[1]. Although it is well known that the results of in vitro tests and cell-based assays are not correlated, i.e., there is a problem with the accuracy of the evaluation, the causes are complex, and many aspects remain unknown.

Disease modeling has actively used iPSCs in drug discovery-related research[2–4]. In a previous study, iPSC-derived neurons with familial and sporadic Alzheimer's disease (AD) were generated[5]. Although AD is classified into different subtypes, it is possible to

[1]Graduate School of Engineering, Muroran Institute of Technology, Hokkaido, Japan. [2]Regenerative Medicine and Cell Therapy Laboratories, Kaneka Corporation, Kobe, Japan. [3]Department of Chemical Science and Engineering, National Institute of Technology, Miyakonojo College, Miyakonojo, Japan. [4]Department of Veterinary Medicine, Research Center of Global Agromedicine, Obihiro University of Agriculture and Veterinary Medicine, Obihiro, Japan. [5]These authors contributed equally: Naoki Nishishita, Kiyotaka Tokuraku. ✉e-mail: Nishishita@kaneka.co.jp; tokuraku@muroran-it.ac.jp

develop subtype-specific AD drugs using AD patient-specific iPSCs[6]. However, in a bid to select candidate compounds, it is challenging to apply an iPSC-based assay system for screening drugs in multifactorial diseases such as AD to accommodate genetic and environmental factors since drug development is labor- and time-intensive. By using the supernatant of a cell culture, it is possible to imitate an environment that is closer to that of a living body, as one example, phosphate-buffered saline (PBS), making it easier to discover more effective compounds.

AD, one of the most well-known dementia-related diseases, exhibits amyloidosis[7]. AD accounts for the majority of dementia-causing diseases[8–10]. The amyloid cascade hypothesis provides the most promising mechanism for the onset of AD[11,12]. In this hypothesis, abnormal aggregation and the accumulation of amyloid β (Aβ) cause neurodegeneration in the brain. Aβ molecules begin to aggregate before the onset of AD and exhibit cytotoxicity in neurons[12–14]. Previously, we reported a real-time imaging method to detect Aβ aggregation using quantum dots (QDs), which are fluorescent semiconductor nanoparticles[15]. This QD-based imaging method was applied to observe the aggregation of other amyloid proteins, tau, α-synuclein, and serum amyloid A (SAA)[16,17]. Moreover, using this method, we developed a Microliter-Scale High-Throughput Screening (MSHTS) system for screening Aβ aggregation inhibitors[18]. The MSHTS system is performed using PBS, which is different from the in vivo environment, except for salt concentration. Separately, living cells contain various proteins, lipids, trace elements, small molecules, and other compounds. Therefore, the identification of materials that exhibit high Aβ aggregate inhibitory activity in the body requires an evaluation of the

activity of candidate substances in vivo. However, the complexity of Aβ's features in living organisms, which contain various biomolecules, causes a dissociation between in vivo and in vitro findings[19]. To resolve the discrepancy between the results derived from an evaluation that uses in vitro and in vivo assays, in this study, we propose a technique that utilizes the culture supernatant of iPSCs, which is a by-product produced by disease modeling in a dish, for screening using the MSHTS system.

In this study, we succeed in the visualization of the Aβ aggregation process in the conditioned media of neurons induced to differentiate iPSCs derived from a male AD patient (i.e., AD-iNeuron) in real-time and evaluation of the aggregation inhibitory activity. We apply this imaging method to develop a High-throughput screening technology for Aggregation Inhibitors of Diseased cell-derived Aggregative Proteins (HaiDap) system (Fig. 1), which uses the supernatant of the neurobasal medium, and used it to evaluate 22 plant extracts. Furthermore, we demonstrate that the HaiDap system could be used to screen materials that prevent the aggregation of various amyloid proteins, tau, α-synuclein, and SAA, which are involved in the pathogenesis of AD, Parkinson's disease (PD), and AA amyloidosis. These results will be useful for highly efficient, accurate, and low-cost screening of disease- and patient-specific drugs involved in protein aggregation.

## Results

### Examination of Aβ aggregation conditions in the culture supernatant of AD-iNeuron

First, we examined whether it was possible to observe Aβ aggregation by QD-labeled Aβ (QDAβ) in the medium for differentiating iPSCs into

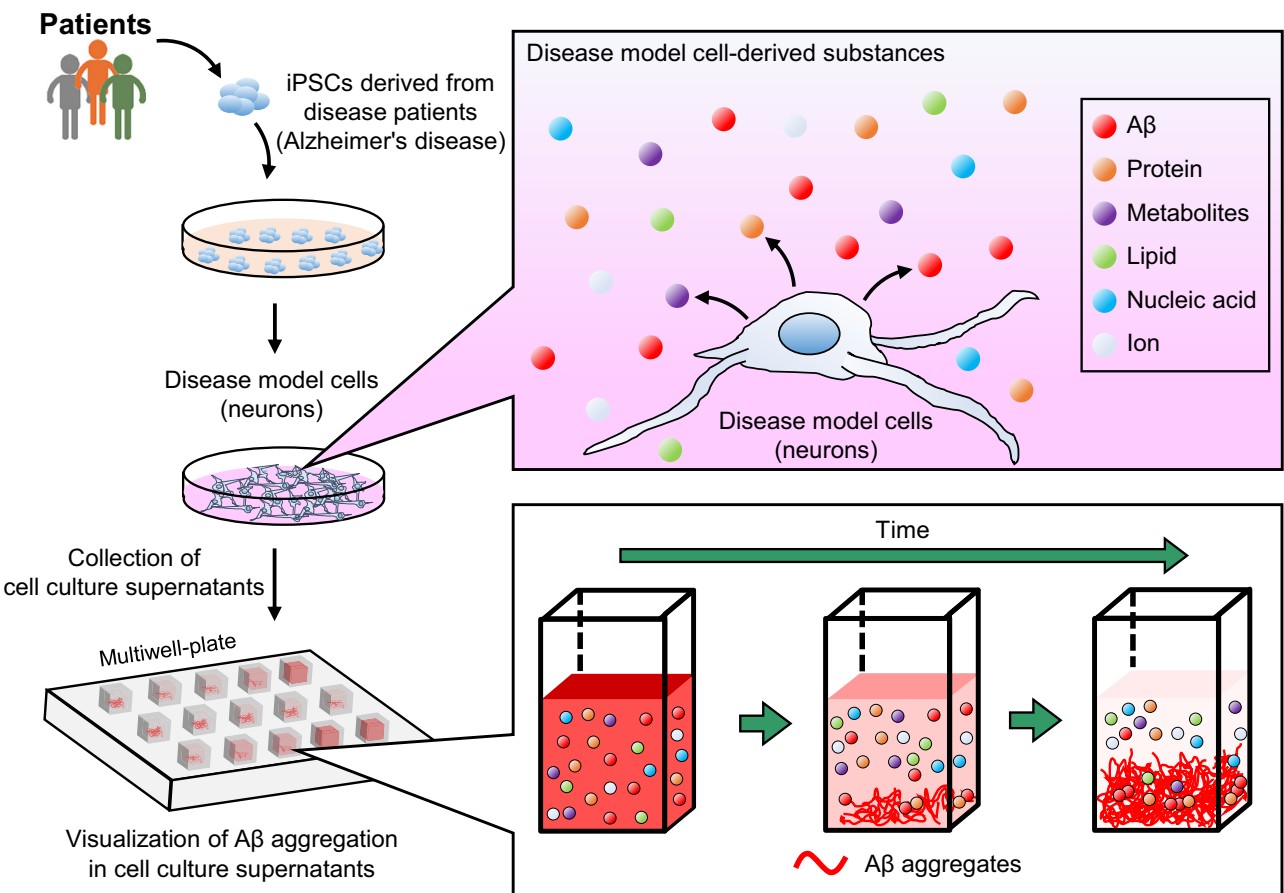

**Fig. 1 | Schematic visualization of Aβ aggregation using the culture supernatant of disease model cells derived from iPSCs.** A culture supernatant containing cell-derived proteins, metabolites, lipids, nucleic acids, and ions is used as a solvent to reproduce the in vivo environment. High-throughput evaluation of the inhibitory effects of various candidate materials on Aβ aggregation is carried out using the culture supernatant.

neurons. In two types of media, a Neurobasal medium and a DMEM/F12 medium with 0.5% B27 solution[20], no aggregates formed (Fig. 2a). The standard deviation (SD) values, which represent the pixel-by-pixel brightness variance of the fluorescence images that are correlated with the amount of Aβ aggregates[18], revealed that B27 significantly inhibited aggregation (Fig. 2b). B-27 strongly inhibited Aβ aggregation in PBS (Supplementary Fig. 1A). We confirmed that 0.4-10% B-27 decreased the SD values, indicating that the amount of Aβ aggregates had decreased (Supplementary Fig. 1B). Although the formation of Aβ aggregates was confirmed in DMEM/F12 medium with 10% FBS, the amount was clearly less (Supplementary Fig. 2a) than that in PBS (Fig. 2a). Since serum albumin, which is found in B27 and FBS, inhibit the polymerization of Aβ[21–23], we attempted to remove albumin from the medium using a 50 kDa filter device (Supplementary Fig. 2b). Aβ aggregates were observed in filtered medium (Supplementary Fig. 2c), suggesting that albumin present in the medium inhibited Aβ aggregation. Furthermore, we investigated the optimal filter size for removing albumin and found that Amicon Ultra, which removes proteins greater than 50 kDa, was the most appropriate filter for our study (Supplementary Fig. 3). Even when media containing B27 were filtered using this filter setting, Aβ aggregates formed (Fig. 2c). Next, we attempted to establish AD-iNeuron (Supplementary Fig. 4) and observed Aβ in the culture supernatant (Fig. 2d, Table 1). In this case, two media, with or without the culture of AD-iNeuron, were used, resulting in a neurobasal medium and in a supernatant of the neurobasal medium based on AD-iNeuron, which were named NbM and NbM Sup, respectively. As expected, in NbM Sup from which albumin had been removed by filtration (NbM Sup Alb(−)) (Fig. 2e), Aβ aggregation was observed (Fig. 2f). We also assessed the composition of the culture supernatant of AD-iNeuron, as well as the concentration of cell-secreted Aβ, metabolites, and electrolytes in the medium (Supplementary Fig. 4, Table 1). The results revealed that Aβ concentration in the culture medium of AD-iNeuron increased over culture time. We previously demonstrated that 79.3% of the Aβ secreted from AD-iNeuron into NbM Sup remained after filtration using a 50 kDa filter device (see US Patent App. 18/285,090, 2024)[24], suggesting that the majority of Aβ secreted from AD-iNeuron remains in NbM Sup Alb(−). To verify whether Aβ fibrils formed in NbM Alb(−) and NbM Sup Alb(−), transmission electron microscopy (TEM) was performed (Fig. 2g). Fibril formation and the incorporation of QDAβ were observed in all conditions, as had been observed in our previous findings[15]. These results suggest that Aβ aggregation can be visualized in the culture supernatant by removing albumin from the medium by filtration.

## Aggregation of various disease-associated amyloid proteins in the culture supernatant of AD-iNeuron

To date, we have observed the aggregation of various amyloid proteins, Aβ, tau, α-synuclein, and SAA, using QDs[16,17,25]. In this study, the aggregation of various amyloid proteins in NbM Sup Alb(−) was observed using QD imaging methods using the same protein concentration that was employed in our previous studies[16,17,25] (Fig. 3). Tau aggregated slightly after 192 h (Fig. 3 tau). We previously reported that heparin and dithiothreitol (DTT) are required to observe Tau aggregation[25]. The binding affinity between heparin and Tau was strong in the presence of DTT, resulting in their fibrillization under reducing conditions[26]. The addition of heparin and DTT may accelerate tau aggregation in this condition. SAA aggregated in NbM Sup Alb(−) (Fig. 3 SAA), as well as in PBS, but the morphology of the aggregates was slightly different[17]. Interestingly, α-synuclein increased the rate of aggregation in NbM Sup Alb(−) (Fig. 3 α-synuclein) compared to aggregation in PBS[16]. Aβ plaques are involved in α-synuclein and Tau seeding in the mouse brain[27]. The culture supernatant of AD-iNeuron contained just over 40 nM of Aβ42 (Supplementary Fig. 4). As mentioned above, it is estimated that ~80% of the Aβ remained in NbM Sup Alb(−) after filtration, and it presumably affected the aggregation of α-

synuclein. Elucidating the mechanism underlying the increased rate of synuclein aggregation in NbM Sup Alb(−) will provide important insight into the pathological elucidation of many proteinopathies, including PD.

## Kinetics of Aβ aggregation in Aβ-containing culture supernatant of AD-iNeuron

When Aβ aggregation in each medium without albumin was observed over time, it was noticed that this process began earlier in NbM Sup (-) than in PBS and NbM Alb(−) [PBS: ~4 h, NbM Alb(−): ~6 h, NbM Sup Alb(−): ~2 h] (Fig. 4a). A logistic regression analysis of Aβ aggregation using the SD value of each image was then performed (Fig. 4b and Table 2). The difference in slope for each sample is caused by different liquid components. The difference between PBS and NbM Alb(−) is mainly due to differences in components such as ions and sugar, whereas the difference between NbM Alb(−) and NbM Sup Alb(−) is due to a difference in cell secretory components. As shown in Table 2, the slope of NbM Sup Alb(−) is higher than that of NbM Alb(−), which might be due to low molecular weight cell-secreted components, such as remaining Aβ in the filtrate, that are widely involved in forming Aβ aggregates within a short period of time. In addition, to confirm whether incubation of the medium at 37 °C affects Aβ aggregation, we prepared NbM medium without AD-iNeuron, and incubated it overnight at 37 °C. Observation of Aβ aggregates in NbM Alb(−) after incubation showed that their morphology was similar to that before incubation (Supplementary Fig. 5a). Moreover, the SD value of brightness did not differ between − incubation and + incubation (Supplementary Fig. 5b). These results suggest that the presence of small molecules or compounds, including Aβ42 secreted from patient-derived iPSCs in the culture supernatant, can accelerate the formation of aggregates and that the QDAβ-based observation methods could detect a variety of Aβ aggregation kinetics. To confirm whether the Aβ in the culture medium can accelerate aggregations, we performed real-time imaging with or without 25 pM of monomer Aβ42 or 25 pM of oligomer Aβ42 in NbM Alb(−) (Supplementary Fig. 6). These concentrations mimicked the concentration of Aβ42 secreted from AD-iNeuron and remaining after filtration. Interestingly, we confirmed that the addition of Aβ oligomer significantly accelerates the aggregation process (Supplementary Fig. 6a and b). On the other hand, the morphology of Aβ aggregates formed when oligomers were added (Supplementary Fig. 6a and 6c) was different from that in NbM Alb(−) (Fig. 4a and c), suggesting that various molecules secreted from the AD-iNeuron also affect aggregation. The 2D images of the aggregates revealed that the shape of Aβ aggregates differed in each medium, with or without cultured AD-iNeuron. Therefore, morphological differences were investigated as three-dimensional (3D) observations (Fig. 4c). The 3D observations of Aβ aggregation in each medium with a confocal microscope revealed that the manner of deposition of aggregates was dramatically different in each medium. Although aggregated particles formed in NbM Alb(−), they were densely packed, and a very thin aggregate structure resulted. In contrast, in NbM Sup Alb(−), aggregated particles were finer than in NbM Alb(−), and they piled up three-dimensionally. As described above, AD-iNeuron-derived NbM Sup Alb(−) contained cellular Aβ42 as an impurity, suggesting that this reflects Aβ aggregation in vivo. To further verify this, we prepared healthy control (HC)-iNeuron-derived NbM Sup Alb(−) and evaluated the kinetics of Aβ aggregation in these samples (Supplementary Fig. 7). The chemical compositions of HC-iNeuron-derived NbM Sup are displayed in Table 1. The most notable differences in the composition of the HC-iNeuron-derived NbM Sup and AD-iNeuron-derived NbN Sup used in this experiment were the concentrations of Aβ40 and Aβ42, both of which were lower in HC-iNeuron-derived NbM Sup (Table 1). Real time imaging showed Aβ aggregation proceeded more rapidly in AD- than in HC-iNeuron-derived NbM Sup Alb(−) (Supplementary Fig. 7a and b). These results supported that the aggregation of 25 μM

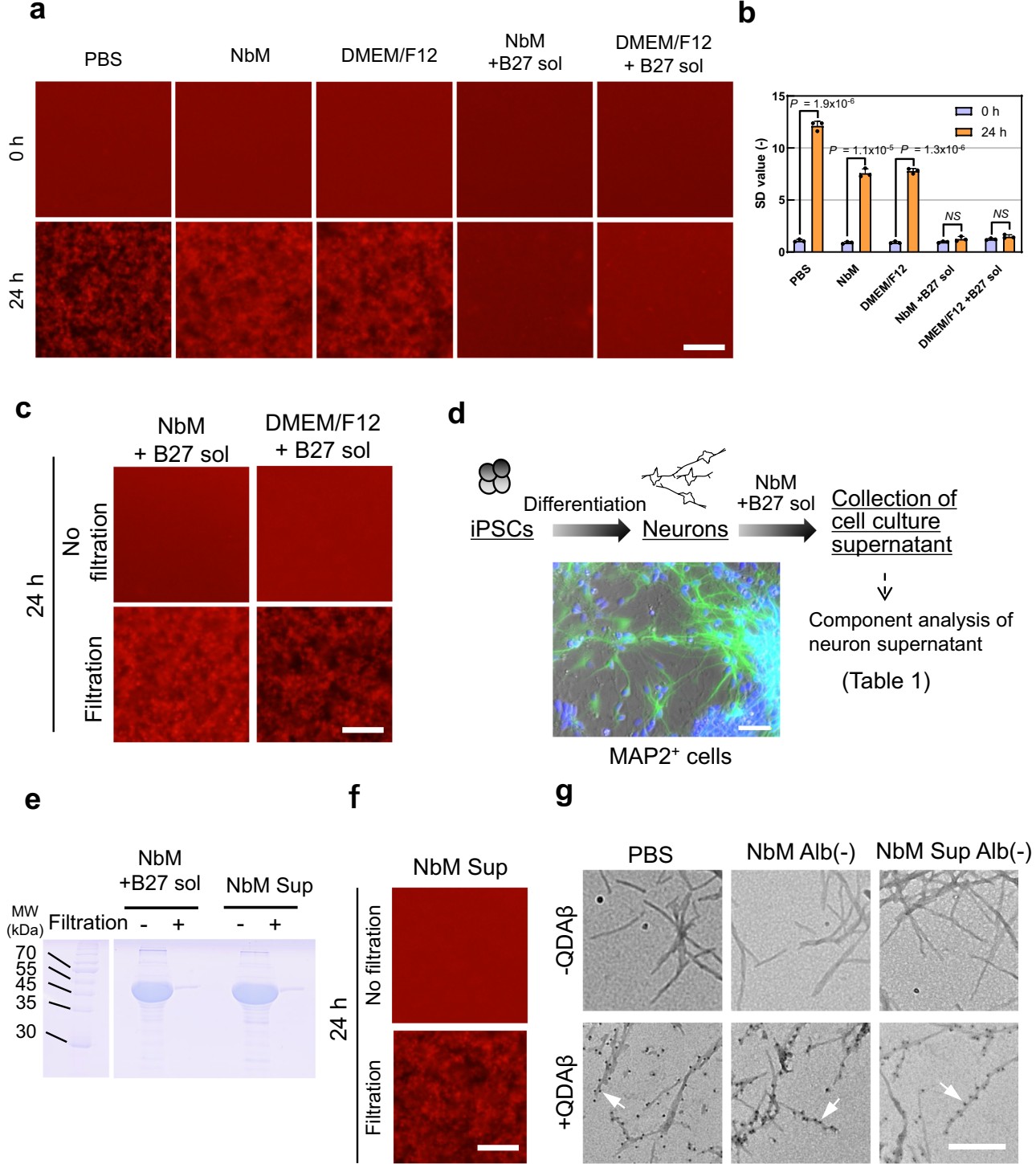

**Fig. 2 | Observation of Aβ aggregates in the culture medium of AD-iNeuron.**
**a** 2D imaging of Aβ aggregates in a 1536-well plate. 25 μM Aβ$_{42}$ and 25 nM QDAβ were incubated in PBS, NbM, and DMEM/F12 medium supplemented with or without 0.5% B27 solution. Bar = 100 μm. **b** Comparison of SD values between 0 and 24 h images in panel a. Note that Aβ aggregates were not formed in the condition with B27 solution. Error bars represent ± SDs of the mean values from fluorescence intensities ($n$ = 3 separate experiments, $p$ < 0.05, Two-sided student's $t$-test). **c** 2D imaging of Aβ aggregates in NbM and DMEM/F12 medium, including B27 solution with or without filtration using a 50 kDa Amicon Ultra filter unit. Note that Aβ aggregates were formed in the filtered condition. **d** Scheme of collection of AD-

iNeuron supernatant. Induced neurons were stained with MAP2 and DAPI (blue). Bar = 50 μm. **e** Representative SDS-PAGE gel (Obtained from two independent test) of Neurobasal medium including B-27 solution (NbM +B27 sol) and NbM Sup with or without filtration using a 50 kDa Amicon Ultra filter unit. **f** 2D imaging of Aβ aggregates in non-filtered supernatant NbM (NbM Sup) or filtered supernatant NbM (NbM Sup Alb(−)). Bar = 100 μm. **g** TEM observations (Obtained from seven independent test) of various amyloid fibrils of 25 μM Aβ without (top) or with (bottom) QDs in indicated solution. Note that QD nanoprobes, observed as black dots indicated by white arrows, bind to fibrils. Bar = 250 nm. All images in panels a and b were captured using a conventional fluorescence microscope. Bar = 100 μm.

**Table 1 | Chemical compositions of indicated solutions**

| Chemical composition | Conc. | PBS | NbM Alb (-) | NbM Sup (AD) | NbM Sup (HC) | Measurement method |
|---|---|---|---|---|---|---|
| pH | | 7.29 | 7.51 | 7.76 | 7.41 | HORIBA |
| Glucose | mmol/L | 0.00 | 23.86 | 17.46 | 21.70 | BF-7 |
| Lactate | mmol/L | 0.00 | 0.00 | 9.94 | 6.10 | BF-7 |
| Glutamine | mmol/L | 0.00 | 0.00 | 0.15 | 0.11 | BF-7 |
| $NH_4^+$ | mmol/L | 0.08 | 0.11 | 0.24 | 0.30 | FLEX2 |
| $Na^+$ | mmol/L | 165.27 | 86.43 | 87.30 | 84.70 | FLEX2 |
| $K^+$ | mmol/L | 1.14 | 5.57 | 5.62 | 5.31 | FLEX2 |
| $Ca^{2+}$ | mmol/L | 0.03 | 1.53 | 1.56 | 1.37 | FLEX2 |
| cell-derived $A\beta_{40}$ [a] | pmol/L | D.L | D.L | 164.68 | 20.82 | ELISA |
| cell-derived $A\beta_{42}$ [b] | pmol/L | D.L | D.L | 43.21 | 11.30 | ELISA |

3 or over, Average.
[a]Detection Limit 0.134 pmol/L.
[b]Detection Limit 0.095 pmol/L.

Aβ was affected by iNeuron-derived Aβ and its oligomers contained in the culture medium at the pM level (Supplementary Fig. 6a and b), which promoted aggregation. Intriguingly, the morphology of Aβ aggregates in HC-iNeuron-derived NbM Sup Alb(−) (Supplementary Fig. 7a bottom) was similar to that in PBS control (Fig. 4a top) but different/dissimilar to those in AD-iNeuron-derived NbM Sup Alb(−) (Fig. 4a bottom). These results indicate that differences in culture medium components other than Aβ also affect aggregation, suggesting that the use of patient-derived iPSC culture supernatants may enable the screening of disease- or patient-specific drugs.

## Automation of the HaiDap system and screening of plant extracts

We next endeavored to establish an automated MSHTS system[25] for Aβ aggregation inhibitors in NbM Sup Alb(−) (Supplementary Fig. 8). In automated MSHTS, the variation in brightness of each pixel in the central part of the image, i.e., the SD value, was used as an index of the amount of aggregates[25]. To accurately calculate SD values, histogram data needs to be normally distributed. As shown in Supplementary Fig. 9a and b, the SD value of each boxed region was measured. If the selection range is too wide or too narrow, the SD value cannot be measured accurately, so the same measurement range, as was suggested in our previous report[25], was adopted. Rosmarinic acid (RA), a polyphenol, is a popular inhibitor of Aβ aggregation that decreases Aβ deposition in AD model mice[28–30]. Thus far, RA has been used as a positive control in MSHTS using PBS. Therefore, we tested whether the Aβ aggregation inhibitory effects of RA could be evaluated, even in NbM Sup Alb(−), by automated MSHTS (Fig. 5a and b). When the concentration of RA was 60 and 300 μM, Aβ aggregation in NbM Sup Alb(−) was inhibited, similar to the PBS results. In addition, after creating an inhibition curve, the $EC_{50}$ value was calculated (PBS: 20.1 ± 2.2 μM; NbM Sup Alb(−): 44.3 ± 8.8 μM). Interestingly, the inhibitory activity of RA against Aβ aggregation in PBS decreased to less than half in NbM Sup Alb(−), indicating that the evaluation results change depending on the composition of the solvent used. On the basis of these results, we developed the HaiDap system, as indicated in Fig. 6. Briefly, iPSCs derived from AD patients differentiated to neuron cells, which are AD model cells. Then, culture supernatants of disease model cells, including extracellular Aβ, were collected. Excessive albumin was removed using a filtering device. Various concentrations of candidate inhibitors were incubated with 25 nM QDAβ and 25 μM $A\beta_{42}$ in a 1536-well plate at 37 °C for 24 h. The aggregation images of each well of the 1536-well plate were captured using an automatic imaging system (Supplementary Fig. 10). After observation, the SD values of fluorescence intensity in each image were analyzed. $EC_{50}$ values of the inhibitors were estimated from inhibition curves.

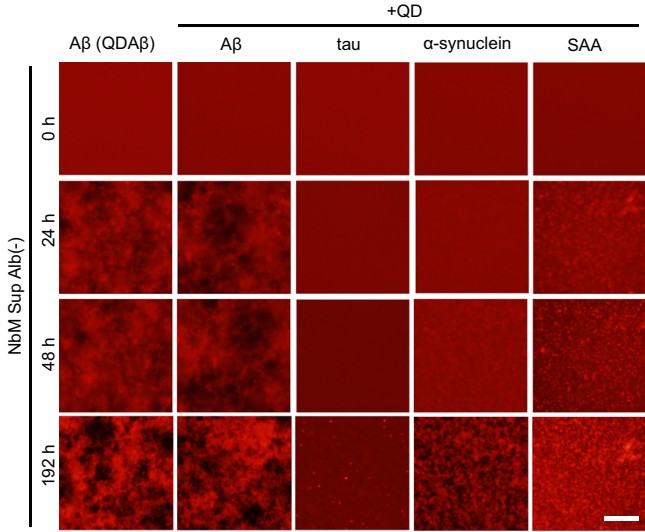

**Fig. 3 | Observation of various amyloid protein aggregates in the culture supernatant of AD-iNeuron.** Real-time imaging of aggregation processes of various amyloid proteins. 25 μM Aβ, 10 μM tau, 6 μM α-synuclein, and 50 μM SAA were incubated with QDs in NbM Sup Alb(−). Aβ and QDAβ (leftmost images) were incubated as a positive control. n = 3 separate experiments. Bar = 100 μm.

Using the HaiDap system, we evaluated the Aβ aggregation inhibitory activity of 22 plant extracts (Supplementary Fig. 11, Supplementary Table 1). The plants that were selected have been used in human diets and are relatively safe for long-term consumption. *Orthosiphon aristatus* exhibited high Aβ aggregation inhibitory activity in NbM Sup Alb(−) (1/$EC_{50}$ value in the HaiDap system: 76.938 ± 7.588 mL/mg). We then compared the Aβ aggregation inhibitory activity of *Hydrangea macrophylla* var. *thunbergia* and *O. aristatus* between the previous MSHTS system (in PBS) and the HaiDap system (Fig. 7a and b). Interestingly, Aβ aggregation inhibitory activity of *H. macrophylla* was not detected by the HaiDap system, and only by the MSHTS system (1/$EC_{50}$ value in the MSHTS system: 11.572 mL/mg). Although *O. aristatus* also showed Aβ aggregation inhibitory activity when using the MSHTS system (1/$EC_{50}$ value = 13.507 mL/mg), the activity obtained with the MSHTS system was less than that obtained with the HaiDap system (1/$EC_{50}$ value = 76.938 mL/mg). These results indicate that the inhibitory activity in PBS and in NbM Sup Alb(−) differed. In other words, the efficacy of Aβ aggregation was inhibited depending on the nature of the solvent.

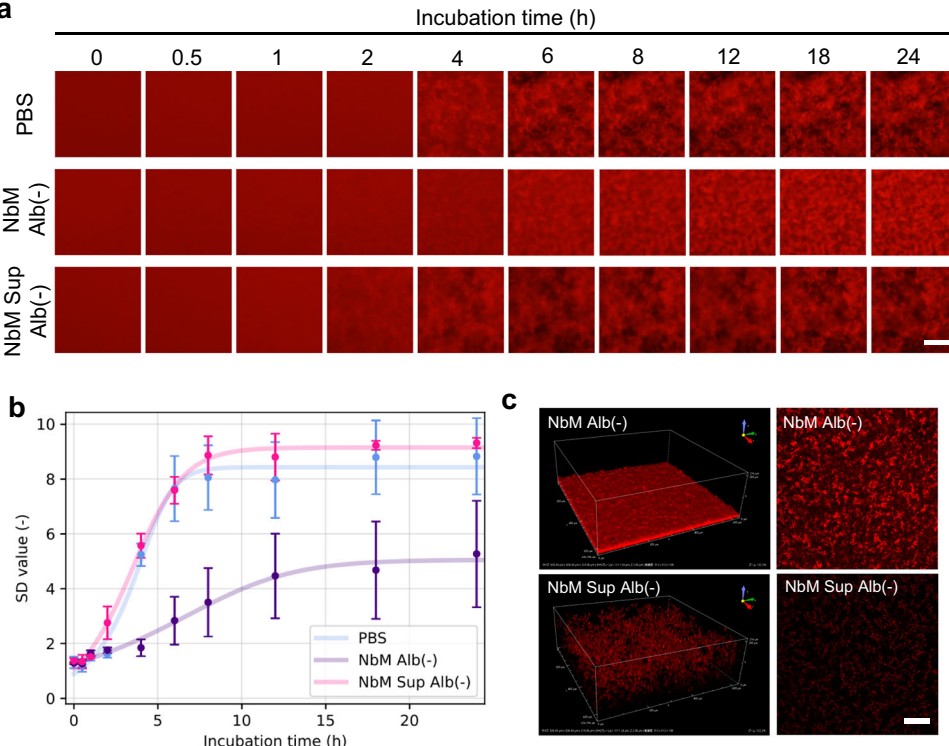

**Fig. 4 | Kinetics of Aβ aggregation in several media. a** Real-time imaging of Aβ aggregation processes in the indicated media. All images were captured using a conventional fluorescence microscope. Bar = 100 μm. **b** Increase of SD values in each condition. SD values were determined by ImageJ software using the 2D images of Aβ. Error bars represent ± SDs of the mean values from fluorescence intensities

($n = 3$ separate experiments). **c** 3D reconstruction images and slice images of Aβ aggregates in NbM Alb(−) and NbM Sup Alb(−) at 24 h of panel a. Note that aggregate sizes and densities are different among indicated media. All images were captured using a confocal laser microscope. Scale bar = 100 μm.

**Table 2 | Logistics analysis of Aβ aggregation**

|             | α    | β    | γ    | θ    | r    |
|-------------|------|------|------|------|------|
| PBS         | 8.43 | 2.57 | 1.18 | 4.07 | 1.70 |
| NbM Alb(−)  | 5.05 | 2.43 | 0.36 | 6.84 | 0.32 |
| NbM Sup Alb(−) | 9.15 | 1.28 | 0.68 | 3.54 | 1.43 |

**Validation of the HaiDap system by iPSC-based screening**

Finally, we verified the efficacy of Aβ aggregation inhibitory activity evaluated by the HaiDap system using iPSC-based screening (Fig. 7c). To measure the Aβ aggregation-suppressing effect of the plant extracts on AD-iNeuron, the neurons were cultured in a Neurobasal medium supplemented separately with the extract of either *H. macrophylla*, *O. aristatus*, *S. aromaticum*, or *G. yesoense* (Supplementary Fig. 12). Aβ aggregation on the surface of neurons was visualized using QDAβ[31], and then Aβ counts, Aβ aggregation size and Aβ intensity were measured using the IncuCyte Live Imaging System (Fig. 7d and Supplementary Fig. 13). In the DMSO control sample, the amount of Aβ aggregation increased around neurons. RA, which showed high activity in the MSTHS system using PBS (Fig. 5a, PBS), showed low aggregation inhibitory activity in the HaiDap system (Fig. 5a, NbM Sup Alb(−)) and iPSC-based assay (Fig. 7d and Supplementary Fig. 13). Furthermore, the *H. macrophylla* extract that showed no detectable activity in the Hai-Dap system also failed to inhibit Aβ aggregation in the iPSC-based assay (Fig. 7d and Supplementary Fig. 13). In conclusion, the extracts of three plants (*O. aristatus*, *S. aromaticum*, and *G. yesoense*) exhibited Aβ aggregation-inhibitory activity in both the HaiDap system and the iPSC-based assay. On the other hand, the results of cell-based assays are also strongly influenced by the cytotoxicity of the samples. To

address this possible concern, we evaluated the cytotoxicity of the three plant extracts used in the experiments (Supplementary Fig. 14a). We then performed a dose-dependent iPSC-based assay using *S. aromaticum* extract, which showed no cytotoxicity even at the maximum concentration used in this experiment, i.e., 500 μg/mL (Supplementary Fig. 14a). The *S. aromaticum* extract showed a dose-dependent response in the iPSC-based assay (Supplementary Fig. 14b and c). The results of the iPSC-based assay were in good agreement with those of the HaiDap system. Although RA was effective in both the MSHTS and HaiDap systems, its effect in the cell-based assay was limited, which left us concerned about the reproducibility of the HaiDap system. Therefore, we searched for a positive control that showed significant effects in all three (MSHTS, HaiDap, and cell-based) assays, and found that epigallocatechin gallate (EGCG) was the appropriate control (Supplementary Fig. 15). EGCG is a catechin found in a variety of plants, including green tea and black tea, and has been reported to be an effective ingredient in inhibiting Aβ aggregation[32,33]. As shown in Supplementary Fig. 15a and b, EGCG could inhibit Aβ aggregation in both of MSHTS and HaiDap systems. Further, EGCG exhibited Aβ aggregation-suppressing effect in a dose-dependent manner in the iPSC-based assay (Supplementary Fig. 15c). These results indicate that EGCG can be a positive control to inhibit Aβ aggregation across different evaluation systems. We thus succeeded in developing a method for more effectively screening Aβ aggregation inhibitory activity in an environment closer to that in living cells than currently available screening systems.

## Discussion

In this study, we developed a high-throughput screening system for Aβ aggregation inhibitors using iPSCs culture supernatant that bridges

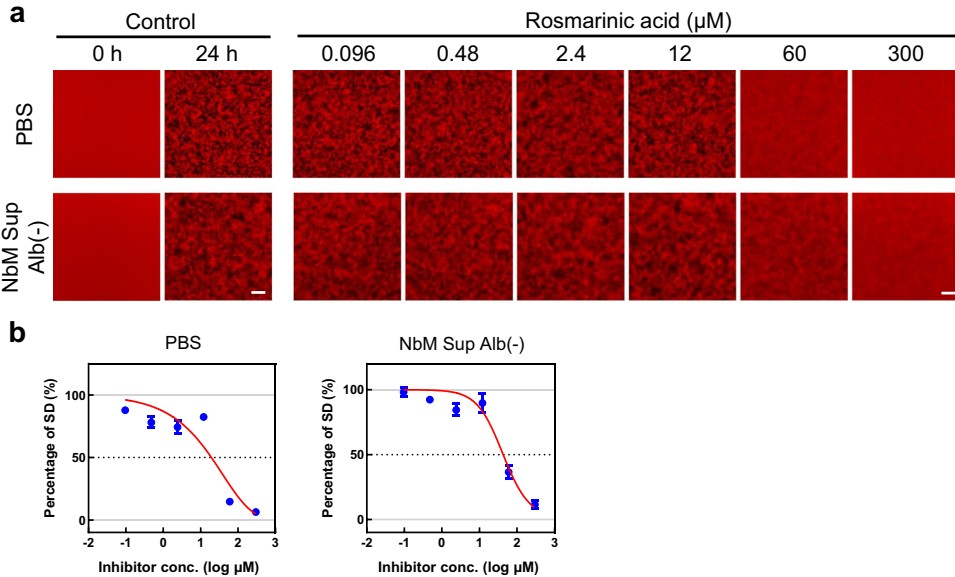

**Fig. 5 | Evaluation of Aβ aggregation inhibitory activity in PBS and NbM Sup Alb(−). a** MSHTS assay in PBS and NbM Sup Alb(−). 25 nM QDAβ$_{40}$ and 25 μM Aβ$_{42}$ were incubated with six concentrations of rosmarinic acid at 37 °C for 24 h. **b** Estimations of EC$_{50}$ of rosmarinic acid. Error bars represent ± SDs of the mean values from fluorescence intensities ($n$ = 4 separate experiments).

in vitro screening and the iPSCs-based assay. Our findings revealed that albumin, which is present in the culture medium used to culture AD-iNeuron, inhibits Aβ aggregation. Many research groups have reported that albumin prevents Aβ polymerization and fibrillation[23,34–36]. Thus, the presence of albumin in blood might be an important anti-AD factor. Indeed, it has been reported that a low concentration of serum albumin increases the risk of AD dementia due to the increased accumulation of Aβ[37]. Albumin contained in the AD-iNeuron culture supernatant has strong Aβ aggregation inhibitory activity and may mask the activity of candidate substances, resulting in reduced sensitivity when detecting inhibition activity. Therefore, albumin was removed from the culture supernatant of AD-iNeuron used in the HaiDap system while leaving most of the low molecular weight biomolecules, such as Aβ (approximately 80% of Aβ$_{42}$ remained) secreted from cells. Consequently, it is now possible to evaluate Aβ aggregation inhibitory activity in the iPSC culture supernatant.

The greatest advantage of the HaiDap system is that it can be used to evaluate Aβ aggregation inhibitory activity of a wide variety of materials with high-throughput in conditions that are very similar to those of a cell-based assay using patient-derived iPSCs. Using an automated system, it is possible to test about 64 samples in the culture medium supernatant at any one time[25]. Generally, an expensive medium and supplementation are required for culturing and maintaining iPSCs over a long-time span. It is thus impossible to test all materials by a cell-based assay using iPSCs due to cost and time constraints. These limitations also apply to experiments using model animals such as mice, rats, etc., making them even more difficult than cell-based assays. In fact, we completed a cost comparison of the popular ThT system using a standard 96-well plate with the HaiDap system and found that the HaiDap system was at least 20 times more cost-effective than the ThT system (Supplementary Fig. 16). In this study, we used the supernatant from the culture of neurons to mimic AD because the main purpose was to search for Aβ aggregation inhibitory material. However, we expect that the HaiDap system could also be applied to discover inhibitory materials of other aggregative proteins. We already succeeded in observing the aggregation of tau, SAA, and α-synuclein, as well as the inhibition of their aggregation, using RA in PBS[16,17,25]. As shown in Fig. 3, we confirmed that SAA and synuclein aggregated in

NbM Sup Alb(−). SAA and α-synuclein are known to cause rheumatism and PD[38–40]. The HaiDap system is thus a powerful tool that has the potential to make a progressive leap forward in the development of various proteinopathy treatments.

The Aβ aggregation inhibitory activity of 22 plant extracts was evaluated using the MSHTS and HaiDap systems, and ultimately nine (40.9% selectivity) and three (13.6% selectivity) plant extracts were selected, respectively (Supplementary Fig. 11, Supplementary Table 1). All three plant extracts (*O. aristatus*, *S. aromaticum*, and *G. yesoense*) that showed Aβ aggregation inhibitory activity in the HaiDap system significantly slowed the rate of Aβ aggregation on the surface of neurons in the iPSC-based assay. Among them, *S. aromaticum*, which did not show any cytotoxicity, also showed dose-dependence in the iPSC-based assay (Supplementary Fig. 14). These results suggest that the HaiDap system is a useful screening method to bridge the gap between in vitro screening and cell-based screening using iPSCs. Incidentally, although RA tended to suppress Aβ deposition in iPSC-based assays (Supplementary Fig. 13), the effect was not as significant as in the MSHTS and HaiDap systems. It was recently reported that Aβ aggregation is suppressed by an increase in monoamines in the brain following the intake of RA[29], and that RA might not directly suppress Aβ aggregation in the brain. These results suggest that RA is not a suitable positive control in all in vitro and in vivo assays. In this study, EGCG showed significant effects in all assays, including MSHTS, the HaiDap system, and the cell-based assay, suggesting that EGCG is more appropriate as a positive control than RA. On the other hand, the EC$_{50}$ value of EGCG for Aβ aggregation inhibitory activity detected by the HaiDap system was low (146.3 μM) (Supplementary Fig. 15a and b), and not comparable to the cell-based assay, where activity was detected at the order of several μM (Supplementary Fig. 15c). This is likely due to some differences between the HaiDap system and the cell-based assay, such as the interaction with the cell membrane and/or cytotoxicity.

It has been confirmed that the AD-iNeuron culture supernatant contains cell-derived Aβ$_{40}$ and Aβ$_{42}$, as well as low molecular weight compounds such as lactate and glutamine (Table 1); however, concerns remain as to the extent to which NbM Sup Alb(−), from which high molecular weight compounds were removed using a 50 kDa filter device, mimics the in vivo environment. In addition, synthetic Aβ

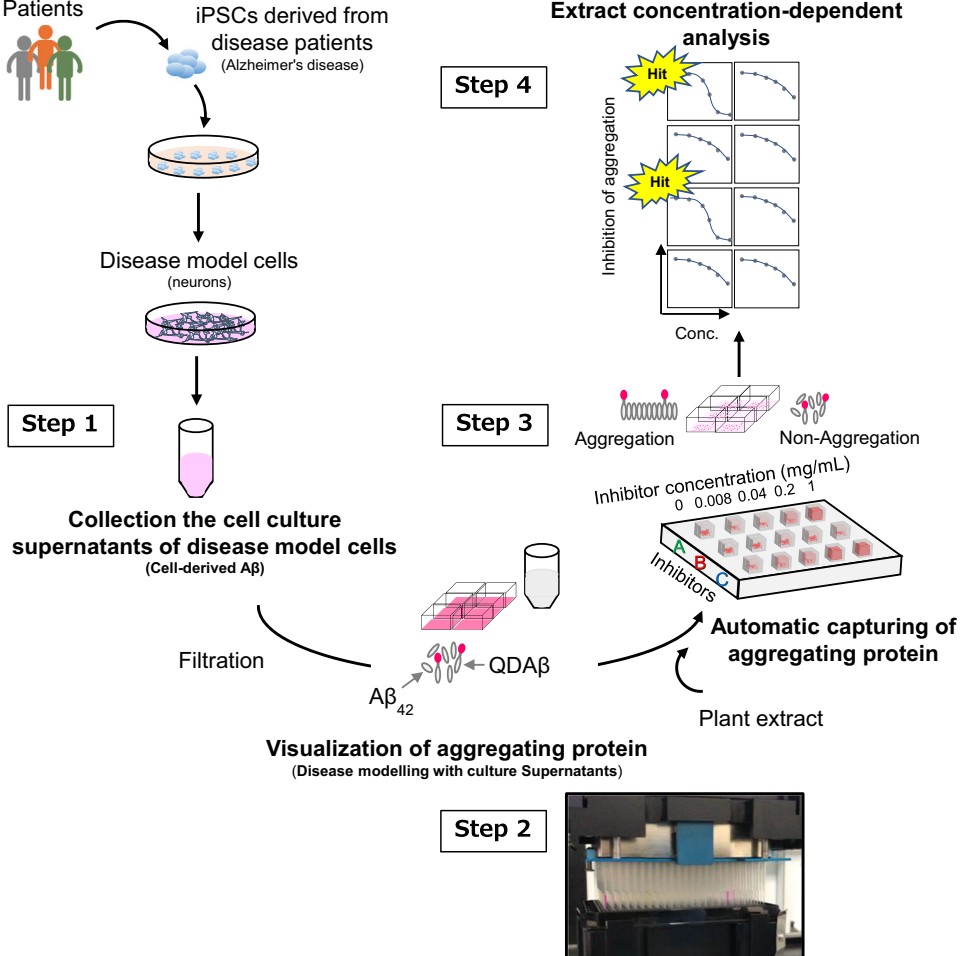

**Fig. 6 | Schematic image of HaiDap system.** Step 1: Collection of the supernatant of model cells (Alzheimer's disease). Step 2: Visualization process of aggregating proteins (including the addition of compounds, etc.). Step 3: Automated imaging of aggregating proteins. Step 4: Calculation of half maximal effective concentration ($EC_{50}$). Calculating the $EC_{50}$ value from the inhibition curve, samples were evaluated for Aβ inhibitory activity. The HaiDap system is a screening strategy that facilitates the selection of compounds and material extracts in supernatants. In the present study, plant extracts were utilized as agglutination suppression samples.

peptides are used in the HaiDap system. To fully mimic in vivo conditions, Aβ peptide extracted from biological samples should be used, but this is very difficult in terms of both time and cost. Synthetic Aβ exhibits neurotoxicity and promotes AD-like pathology in primates[41–43], and is therefore considered to at least mimic AD pathology to some extent. Instead of synthetic Aβ, it is possible to use recombinant Aβ, which allows the preparation of relatively large amounts of Aβ peptide, but this raises concerns about contamination with molecules derived from the host cells used for expression and interactions with these molecules during the expression and purification processes. To accelerate the screening speed and increase the detection sensitivity, we adopted an Aβ concentration of 25 μM in the

MSHTS and HaiDap systems, which also does not completely mimic in vivo conditions. However, perfectly mimicking aggregation, which takes decades, is counter to the need for rapid screening, which is a dilemma for most screening systems. We believe that the substances screened by the HaiDap system are candidates for the next cell-based assays, animal experiments, and human trials, and that the HaiDap system is an effective first screening step that enables rapid evaluation.

Although the HaiDap system enables efficient detection of Aβ aggregation and identification of inhibitors, several limitations must be acknowledged. First, all conditioned-media experiments were performed using a single patient-derived iPSC line (HPS0854, male AD), and therefore, the system's performance cannot be generalized across

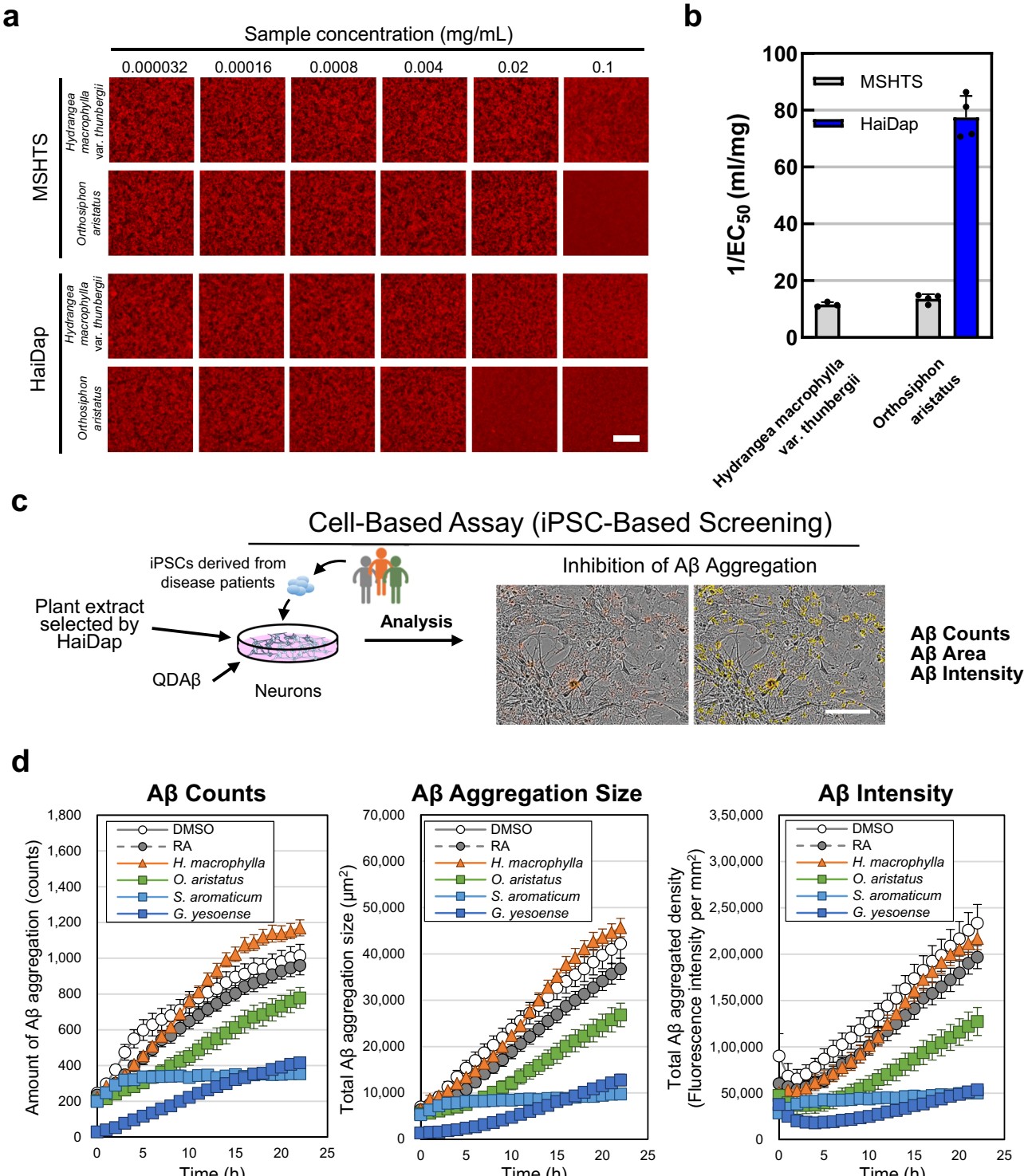

**Fig. 7 | Evaluation of Aβ aggregation inhibitory activity using the HaiDap system and iPSC-based screening. a** 2D image of concentration-dependent inhibition of Aβ aggregation of plant extracts observed by the MSHTS and HaiDap systems. Two samples were evaluated in PBS and NbM Sup Alb(−). Bar = 100 μm. All images were captured using a conventional fluorescence microscope. **b** Aβ aggregation inhibitory activity of indicated samples calculated by the MSHTS and HaiDap systems. Error bars represent ± SDs of the mean values from fluorescence intensities ($n = 4$ separate experiments). **c** Schematic image of cell-based assay (iPSC-based screening). Bar = 200 μm. In this assay neurons co-incubated with 5 μM Aβ$_{42}$, 30 nM QDAβ, and plant extracts were measured by Incucyte every hour. **d** Aβ counts, Aβ aggregation size and total red object integrated intensity, expressed as red calibrated units × μm²/well, were used in the QDAβ method. DMSO is a control experiment that did not contain any extract, and RA is an experiment in which rosmarinic acid was added instead of the extract. Error bars represent ± SEs of the mean values from 9 independent regions.

different donors or disease subtypes; aggregation kinetics and inhibitor responses may vary substantially among patients. Second, while our results indicate that cell-secreted factors accelerate aggregation compared with albumin-depleted basal media, the exact molecular species responsible—whether Aβ oligomers or other metabolites—remain unidentified and will require further biochemical fractionation. Finally, assay-dependent differences were apparent: PBS-based MSHTS and HaiDap showed markedly different selectivity profiles (9/22 vs 3/22), and several compounds (e.g., RA, EGCG) exhibited differing potency depending on the assay matrix, underscoring the need to interpret PBS-based hits cautiously and validate them in conditioned-media and cell-based systems.

In conclusion, we developed the HaiDap system to achieve efficient, accurate, and low-cost drug discovery at the preclinical stage. We demonstrated that this system can also be used to observe the aggregation of amyloidogenic proteins such as α-synuclein in differentiated cell culture supernatants, which contain cell secretions. Recently, propagative α-synuclein seeds have been used as serum biomarkers for synucleinopathies[44]. The HaiDap system is a type of evaluation technology that allows the microliter-scale high-throughput evaluation of drug activity in the presence of disease- and patient-specific secreted substances such as Aβ oligomers and propagative α-synuclein seeds. We believe that the HaiDap system will be used in the future to discover lead compounds that are highly effective for various proteinopathies.

## Methods
### Reagents
Synthetic peptides of human Aβ42 purified by reversed-phase column chromatography (#4349-v; Peptide Institute, purity based on HPLC: 97.4%) and Cys-conjugated Aβ40 (#23519; Anaspec, purity based on HPLC: 97%) were purchased commercially. These synthetic peptides were monomerized using 1,1,1,3,3,3-hexafluoro-2-propanol (#083-04231, Wako Pure Chemical Industries) and stored at −20 °C as previously reported[18]. Aβ oligomerization was performed according to a reported protocol[45], and 1 mM Aβ was diluted with ultra-pure water to 100 μM and incubated for 12 h at 37 °C. The solution was then centrifuged at 16,900 × g for 5 min at 4 °C. The supernatant was collected and stored in a deep freezer (−80 °C).

### Preparation of QDAβ nanoprobes
QDAβ nanoprobes were prepared using QD-PEG-NH2 (Qdot 605 ITK Amino (PEG) Quantum dot; #Q21501MP, Thermo Fisher Scientific) according to our previous reports[18,25], as follows: 10 μM QD-PEG-NH2 was first reacted with 1 mM sulfo-EMCS (#22307; Pierce/Thermo Fisher Scientific) in PBS for 1 h at room temperature. After quenching and eliminating unreacted sulfo-EMCS, the QD-PEG-NH2-bound sulfo-EMCS was reacted with 74 μM of Cys-conjugated Aβ40 for 1 h at room temperature. QDAβ concentration was determined by comparing absorbance at 350 nm to unlabeled QD-PEG-NH2.

### Preparation of tau, α-synuclein, and SAA
Bacterial expression and purification of the tau MBD fragment were carried out as described previously[25]. Briefly, the expression plasmids were transformed into *Escherichia coli* (Rosetta (DE3) pLys, Novagen, Burlington), and plasmid expression was induced by 1 mM isopropyl-1-thio-β-D-galactopyranoside. The heat-stable fraction of each extract was subjected to successive column chromatographies using a Bio-Scale Mini UNOsphere™ S (Bio-Rad) and a TOYOPEARL butyl column (Tosoh). Protein concentration was estimated using the method described by Lowry et al.[46], using BSA as the standard. SDS-PAGE was carried out according to the method of Laemmli et al.[47].

The preparation of α-synuclein was performed as described previously[48], with some modifications. Briefly, Rosetta (DE3) cells harboring pT7-7 asyn WT plasmid (Addgene plasmid #36046;

Watertown) were grown in Luria-Bertani medium supplemented with 100 μg/mL ampicillin and 34 μg/mL chloramphenicol at 37 °C. When OD600 of the culture reached 0.6, the cells were induced with 0.5 mM isopropyl thio-β-D-galactoside for 3 h. Harvested cells were suspended in a buffer (10 mM Tris-HCl (pH 8.0), 1 mM EDTA, and 1 mM phenylmethylsulfonyl fluoride), then lysed by sonication for 10 s. After centrifugation at 10,000 × g, the resultant supernatant was boiled for 20 min and centrifuged once again at 16,000 × g. Ammonium sulfate was gently added to the supernatant until it reached a saturation of 35%. After centrifugation at 13,500 × g, additional ammonium sulfate was added until the supernatant reached 50% saturation. After further centrifuging at 13,500 × g, the pellet was resuspended in 10 mM Tris-HCl (pH 7.4) and dialyzed overnight against 10 mM Tris-HCl (pH 7.4). The dialysate was applied to a HiTrap Q HP column (GE Healthcare Life Sciences), and the protein was eluted using a linear gradient of NaCl (0–500 mM) in 10 mM Tris-HCl (pH 7.4). The fraction containing α-synuclein was desalted using a PD Midi trap G-25 gel filtration column (GE Healthcare Life Sciences). The concentration of α-synuclein was determined using an extinction coefficient of 5600 M⁻¹·cm⁻². Purified α-synuclein was snap-frozen in liquid N2 and stored at −80 °C.

The preparation of mouse SAA (mSAA) was performed as described previously with some modifications. Briefly, mSAA cDNA was inserted into the pET-15b vector, expressed in *E. coli* Rosetta cells and purified[49]. *E. coli* was grown in Luria-Bertani medium supplemented with 25 μg/mL chloramphenicol and 100 μg/mL ampicillin at 37 °C. When OD600 of the culture reached 0.6, cells were induced by 1 mM isopropyl thio-β-D-galactoside for 5 h. After centrifugation at 3000 × g, the pellet was suspended in a 10 mM HEPES buffer (pH 7.4) and washed twice. After washing, the pellet was resuspended in buffer A (10 mM HEPES (pH 7.4), 500 mM NaCl, 0.5% TritonX-100), sonicated (5 kHz) for 5 min, and centrifuged at 38,000 × g for 30 min. The supernatant was applied to a Ni Sepharose™ 6 Fast Flow column (GE Healthcare Life Sciences) then washed with buffer B (10 mM HEPES, 500 mM NaCl, pH 7.4, 0.5% TritonX-100, 50 mM imidazole). mSAA protein was eluted using buffer C (10 mM HEPES (pH 7.4), 100 mM NaCl, 500 mM imidazole), dialyzed for 8 h in 10 mM HEPES buffer, then centrifuged at 27,000 × g for 30 min at 4 °C. The Lowry method was used to determine mSAA protein concentration in the supernatant. The prepared mSAA was diluted with PBS (pH 7.4) to a concentration of 1 mg/mL and stored at −80°C until use.

### Cell culture
In this study, we used one iPS cell line HPS0854 (RIKEN BRC). Information about the cells is provided next. Cell line: HPS0854; classification: iPSCs; gender: male; tissue: skin; disease name: AD. AD-iPSCs were maintained under feeder-free conditions using mTeSR1 medium (Stem Cell Technologies, Vancouver, Canada) on an hES-Matrigel®-coated plate. Medium was changed daily. The passaging method was described previously[50]. Human iPSC-derived neural progenitor cells (NPCs; catalog #200-0620, StemCell Technologies) were generated from the SCTi003-A induced pluripotent stem cell line, which was established from peripheral blood mononuclear cells (PBMCs) obtained from a healthy female donor. These NPCs exhibit multipotency and were maintained in STEMdiff™ Neural Progenitor Medium (catalog #05833, Stem Cell Technologies) according to the manufacturer's instructions.

### Collection of culture supernatant containing cell-derived Aβ
To collect the culture supernatant containing secreted Aβ with AD-iNeuron, 1.8-3.5×10⁵ cells/cm² iPSCs were seeded on a Matrigel (#354277, Corning) coated 6-well plate in mTeSR1 medium (#85850, Stem Cell Technologies) with 10 μM Y-27632 (#18190, Nacalai Tesque). Then, the STEMdiff™ SMADi neural induction kit (#08582, Stem Cell Technologies) was used with the monolayer culture protocol to promote efficient conversion of iPSCs to CNS-type neural progenitor cells (NPCs). Moreover, in order to optimize the subculture and maintenance of the NPCs, STEMdiff™ Neural Progenitor Medium (#05833,

Stem Cell Technologies) was utilized. A total of $1.9–2.3 \times 10^5$ cells/cm² NPCs were seeded on Growth Factor Reduced Matrigel (GFRM)-coated 6-well plates. During passage, Accutase (Innovative Cell Technologies) was used for cell detachment, and the medium was exchanged every three days. NPCs were cryopreserved using STEMdiff™ Neural Induction Medium (#05838, Stem Cell Technologies). To generate induced AD-iNeuron derived from NPCs, $1.0 \times 10^5$ cells/cm² NPCs seeded on GFRM-coated 96-well plates using phenol-red-free Neurobasal® medium (NbM) (Thermo Fisher Scientific) with B27 (×50) (#A1895601, Thermo Fisher Scientific) plus 100 ng/mL of recombinant human sonic hedgehog[51–53] (#78065, Stem Cell Technologies) (Supplemental Fig. 4a). In this AD-iNeuron induction step, glutamine-free Neurobasal medium was used. Cells were incubated at 37 °C in a 5% $CO_2$ atmosphere. One day after plating NPCs and while collecting half the culture supernatant every 3 days, NbM with B27 medium was added and cultured for 21 days to allow for neural differentiation. The changes in cell morphology during each culture process were confirmed by images (Supplemental Fig. 4b). The culture supernatant was centrifuged at $2000 \times g$ for 3 min, and the supernatant was collected in a tube and stored at −20 °C. The generation of healthy control neurons (HC-iNeuron) was described previously[54]. The differentiated HC-iNeuron was cultured for 28 days in NbM supplemented with B27, as in the case of AD-iNeuron. The culture supernatant was centrifuged at $2000 \times g$ for 3 min, and the resulting supernatant was stored at −20 °C.

## Immunocytochemistry

AD-iNeuron was labeled with NeuroFluor™ NeuO (green) (Fig. 2d). The other iPSCs, NPCs, and neurons were also confirmed by immunocytochemistry (Supplemental Fig. 4c). These cells were washed once with D-PBS, then fixed using 4% paraformaldehyde for 30 min at 4 °C. Fixed cells were permeabilized and blocked with PBS containing 0.1% TritonX-100 and 1% BSA for 1 h at room temperature. Samples were incubated with primary antibody diluted in PBS containing 1% BSA at 4 °C overnight. After washing with PBS three times, samples were incubated with secondary antibody diluted in PBS, and cells were observed using a BZ-X710 microscope (Keyence). The antibodies used in this study were: anti-Nanog antibody (dilution ×250, host: rabbit, ab109250), anti-SSEA4 (dilution ×500, host: mouse, #MA1-021), anti-NESTIN (dilution ×150, host: mouse, #A24345), rabbit anti-PAX6 (dilution ×150, host: rabbit, #A24340), anti-SOX2 (dilution ×150, host: rabbit, #ab137358), anti-MAP2 (dilution ×250, host: mouse, #ab254144), Alexa 488 donkey anti-mouse (dilution ×250, #A24350), Alexa 594 donkey anti-rabbit (dilution ×250, #A24343), and Alexa 594 goat anti-mouse (dilution ×100, #A21155). Cell nuclei were stained blue with DAPI (19178-91, Nacalai Tesque).

## Exclusion of albumin using the filtering device

To exclude albumin derived from the B-27 solution, a commercial filtering device, Amicon Ultra (#UFC505096, UFC905024, Millipore), was used (for details, see US Patent App. 18/285,090, 2024). The time and speed of centrifugation were set according to the manufacturer's procedure. Before filtering the medium, the filtering device was cleaned by washing it twice with water in a centrifuge. Then, 0.1% Tween-20 solution was applied to the filtering device. After incubation for 2 h at room temperature, the filtering device was centrifuged three times to remove the 0.1% Tween-20 solution. Finally, the culture supernatant obtained from AD-iNeuron was filtered.

## Aβ aggregation model

The Richards growth curve, which is a generalized formulation of the logistic function, was used to model time evolution of the intensity SD, which represents Aβ aggregation. The growth curve is defined by:

$$f(t, \alpha, \beta, \gamma, \theta) = \frac{\alpha}{[1 + \beta e^{(-\gamma(t-\theta))}]^{1/\beta}} \quad (1)$$

where $\alpha$ is the amplitude and $\gamma$ describes the growth rate. $\beta$ and $\theta$ together represent the growth rate $r(\beta, \theta)$. Here, the growth rate (r) describes the steepest slope and can be written as $r = \max(df/dt)$. Least square optimization was used to fit the curve $f(t, \alpha, \beta, \gamma, \theta)$ to the data and to compute its characteristic properties.

## Analysis of the culture supernatant

Analyses of the collected culture supernatant (on days 1, 7, 14, and 21) were performed using BioProfile® FLEX2™ (Nova Biomedical). The culture supernatant medium was analysed and evaluated for pH, glucose, lactate, and electrolytes (calcium, sodium, potassium). Aβ secreted from neurons was quantitatively analyzed by the Molecular Devices SpectraMax i3X using an ELISA kit (Human βAmyloid (1-42); FUJIFILM 298-62401, Human βAmyloid (1-40) Wako II; FUJIFILM 298-64601).

## Preparation of plant extracts

One g of dry plant material (*Hydrangea macrophylla* var. *thunbergia*, *Orthosiphon aristatu*, *Ocimum americanum*, *Glebionis coronaria*, *Stevia rebaudiana*, *Cycas revoluta*, *Citrus tankan*, *Ipomoea batatas*, *Canavalia gladiata*, *Azadirachta indica*, *Ocimum basilicum*, *Clitoria ternatea*, *Coix lacryma-jobi* var. *mayuen*, *Senna occidentalis*, *Eriobotrya japonica*, *Lablab purpureus*, *Syzygium aromaticum*, *Fagopyrum esculentum*, immature *Pisum sativum* (green), mature *P. sativum* (yellow)) was immersed in 90% ethanol, and stirred at $10 \times g$ at 28 °C for 20 h in the dark to separate the extracts. The parts of each plant used in the experiment are shown in Supplementary Table 1. Extracts were centrifuged ($2000 \times g$) for 20 min and concentrated with an evaporator after filtration. Each extract was transferred to a glass vial, weighed after lyophilization, and dissolved in DMSO to a final concentration of 100 mg/mL each.

## MSHTS and HaiDap systems

The SD values were determined by a modified MSHTS system, which was previously described[18,25]. More specifically, various concentrations of aggregation inhibitors, 25 µM Aβ$_{42}$ and 25 nM QDAβ in PBS or NbM Sup Alb(−), were incubated in a 1536-well plate (#782096, Greiner Bio-One) at 37 °C for 24 h. The QDAβ-Aβ$_{42}$ aggregates that formed in each well were observed by an inverted fluorescence microscope (TE2000, Nikon) equipped with a color CCD camera (DP72, Olympus) and an objective lens (Plan Fluor 4×/0.13 PhL DL, Nikon). SD values of fluorescence intensities of 40,000 pixels (200 × 200 pixels) around the central region of each well were measured by ImageJ software (NIH). EC$_{50}$ values were estimated using Prism software (GraphPad Software).

## Automated MSHTS and HaiDap systems

The automated MSHTS system used was described in our previous report[25]. Briefly, sample preparation was carried out using automated workstation JANUS G3 (Perkin Elmer). To create a series of dilution samples, dilution buffer (10% EtOH, 1×PBS, or NbM Sup Alb(−)) was aspirated using a conductive chip that can detect the level of liquid and inject it into each well of a 384-well plate (#HSP3801, BIO-RAD). Stock sample solutions were injected manually into vacant wells of the 384-well plate. This dilution step was repeated five times so that a six-dilution series was prepared for each sample. The diluted samples in the 384-well plate were aspirated using a 384-chip head and injected into a 1536-well plate. The Aβ solution, which is a mixture of 50 nM QDAβ and 50 µM Aβ$_{42}$, was prepared in a microcentrifuge tube at 4 °C, aspirated by a conductive chip that can detect the level of liquid, and injected into a cold 384-well plate placed at 4 °C. Finally, the Aβ and sample solutions were mixed by pipetting three times into a 1536-well plate. The 1536-well plate was sealed with a plate seal to prevent evaporation. In order to remove bubbles and flatten the liquid surface, the 1536-well plate was centrifuged at $1530 \times g$ for 5 min in a multi-well plate centrifuge (PlateSpin II, KUBOTA). After centrifugation, the 1536-

well plate was observed by an inverted fluorescence microscope system (ECLIPSE Ti-E, Nikon) equipped with a color CMOS camera (DS-Ri2, Nikon). QD was imaged using a 4× objective lens (Plan Apo λ 4×/0.20, Nikon) and a TRITC filter set (TRITC-A-Basic-NTE, Semrock).

## TEM observations

A total of 10 μL of amyloid fibrils was prepared in PBS or NbM Sup Alb(−). Each sample was transferred onto a 200-mesh copper grid (Nisshin) and incubated for 5 min at room temperature. Grids were then washed twice with water. Samples were negatively stained twice with 1% phosphotungstic acid. Specimens were examined under an H-7600 transmission electron microscope (Hitachi) at 60 kV.

## Confocal laser scanning microscopy

Z-stacks images of Aβ aggregates were captured using an inverted microscope (Ti-E) and a confocal laser microscope system (C2 Plus; Nikon) equipped with an objective lens (PlanApo λ 20×/0.75 NA; Nikon). Images were captured and oversampled by taking 2 μm z-steps between the acquired images, which were analyzed with NIS-Elements software (Nikon).

## Measurement of Aβ aggregation in the AD-iNeuron

Aβ aggregation on the cell surface in indicated conditions was measured using the IncuCyte Live Imaging System (Sartorius). Neurons co-incubated with 5 μM Aβ$_{42}$, 30 nM QDAβ, and plant extracts were measured by Incucyte every hour. The Aβ aggregation process (Aβ counts, Aβ aggregation size, and Aβ aggregation integrated intensity), when surrounded by yellow lines, indicates the perforated areas in the cell monolayer. For apoptosis assays, the IncuCyte Caspase 3/7 green dye kit (#4440, Sartorius) was used at a final concentration of 1:1000. Final concentrations of toxic control compounds added to the plate were: 5 μM staurosporin, while the extracts of four plants (*H. macrophylla* var. *thumbergii*, *O. aristatu*, *S. aromaticum*, and *G. yesoense* var. *yesoense*) were tested at five concentrations (0.8, 4, 20, 100, and 500 ng/μL: $n = 20$ samples). Treated plates were incubated at 37 °C for 24 h, imaged at 20× magnification using phase and green channels at nine images per well, and analyzed using IncuCyte software.

## Ethics statement

Human iPSCs and iPSC-derived NPCs used in this study were obtained from tissues collected by a third-party organization after written informed consent had been appropriately obtained from patients and healthy donors. These cells were purchased and used in accordance with all required procedures. All procedures related to the acquisition and handling of human tissues and cells, as well as all associated research protocols, were approved by the Ethics Committee of Kaneka Corporation (Approval No: 2025-5). The research was conducted in compliance with all applicable regulations, including the required Material Transfer Agreements (MTAs) and relevant licensing requirements. Race and ethnicity information was obtained solely from publicly available cell line metadata provided by RIKEN BRC and StemCell Technologies. The study used one iPSC line (HPS0854) derived from a male donor with Alzheimer's disease, and NPCs derived from an iPSC line (SCTi003-A) originating from a healthy female donor. No additional personal or socially relevant information was collected, accessed, or used. Socially sensitive attributes such as race, ethnicity, socioeconomic status, or related characteristics were not used to guide the study design, data analyses, or interpretation.

## Data availability

All data supporting the findings of this study are provided in the manuscript and Supplementary Information file. Source data are provided with this paper as a Source Data file.

## Code availability

Data analysis for Fig. 4b, Supplementary Fig. 6b and Supplementary Fig. 7b was performed as follows: The software application to perform logistic regression and visualization was written in Python. Nonlinear regression was implemented using the least square fitting implementation provided by the Scipy library. The plots were created using matplotlib. The web app, as well as its source code, is available at https://zenodo.org/account/settings/github/repository/StefanBaar/LGR_drug_screening. All libraries used for producing the figure are open-source and are referenced at the provided URL.

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

## Acknowledgements

This work was supported by JSPS KAKENHI Grant Number JP24K08627 (K.T.) and JST Grant Number JPMJPF2213 (to K.T.). TEM observations were conducted at the Institute for Molecular Science, supported by "Advanced Research Infrastructure for Materials and Nanotechnology in Japan (ARIM)" of the Ministry of Education, Culture, Sports, Science and Technology (MEXT), Proposal Number JPMXP1222CT0078.

## Author contributions

M.K. performed major experiments and wrote the manuscript. N.N. and A.K. performed iPSC experiments. K.A., S.B., and S.W. performed kinetic analyses. K.U. prepared plant extracts. TQPN prepared α-synuclein. K.W. prepared SAA. N.N. and K.T. designed the project. All authors have read and agreed to the published version of the manuscript.

## Competing interests

N.N. and A.K. are now employees of Kaneka Corporation. This research was partially funded by Kaneka Corporation. A patent related to the HaiDap system described in this manuscript has been filed. Patent applicant: Kaneka Corporation and Muroran Institute of Technology. Inventor(s): N.N., A.K., K.T., K.U., M.K. Application number: US.202218285090.A and JP2023511699A, Status: published. The remaining authors declare no competing interests.
