## [Transparent Peer Review file · Nature Communications]

A high-throughput conditioned-media-based screening system identifies inhibitors of aggregation induced by iPSC-secreted amyloid β

Corresponding Author: Professor Kiyotaka Tokuraku

Version 0:

Reviewer comments:

Reviewer #1

(Remarks to the Author)

In the manuscript entitled "High-throughput screening technology for aggregation inhibitors of diseased cell-derived aggregative proteins (HaiDap) system," the authors have developed and refined a screening system for amyloid aggregation inhibitors using iPSCs derived from Alzheimer's disease patients. The goal of the paper is to study aggregation inhibitors in a more disease relevant context. This paper reports on the aggregation of A β 42, and other disease-associated amyloid proteins, in supernatant of the differentiated iPSC cell line, in addition to the kinetics of aggregation. The proposed methodology has optimized neural cells derived from an AD patient and reports the aggregation in presence of characterized culture media obtained from AD-derived neural cells. Cleverly, quantum dot labeled Ab40 with unlabeled Ab42 were used to define the shape, morphology and the texture of the aggregates at the scale of several hundreds of micrometers.

The scale to which the authors can screen compounds is impressive, and markedly higher throughput than most assays. I appreciate their efforts to do aggregation studies that are more biologically relevant to pathology, as this is important. However, a number of issues are noted below that in my opinion limit the probability that this system recapitulates key aspects of neurodegenerative disease.

- 1) The model system was established using cells derived from one AD patient (n=1). Beyond the limited n value, background information about the patient is missing, such as the genotype, if the patient was a familial or sporadic case of AD, age of onset, MMSE scores, etc.
- 2) Filtration also removes other proteins in solution than albumin (Fig. 2e), so it is possible that species beyond albumin are removed and likely are relevant to the aggregation kinetics of Ab42 since this is so dependent on solution conditions. While the secretion of species from the AD cells makes this system more relevant than standard immortalized cell lines, it is not known that this reaction mixture actually recapitulates key aspects of AD.
- 3) The concentrations of amyloidogenic proteins used (25 μ M for A β 42) to test for aggregation in NbM sup Alb- medium are quite a bit higher than those used in typical in vitro aggregation studies (purified Ab42 typically aggregates with a half-time of 2 h at a concentration of 2 μ M). One would predict their system would show faster aggregation at such concentrations (eg a few μ M), considering the crowded nature of their media relative to a typical simple buffer system. It is unsurprising that aggregation begins immediately (Fig 7) considering the very high concentration of Ab42 and the crowded nature of the media. As a result, mechanistic insight into the microscopic mechanisms impacted by candidate compounds is limited in this system. Since the purpose of this system is to be more physiologically relevant, using at least 50-100x greater concentrations of Ab42 than are necessary to observe cytotoxicity is an issue.
- 4) The rationale for using plant extracts for screening is not clear. The authors should consider elaborating on the plant extracts that were screened and provide additional background information on the significance of deploying specific plant extracts for this advanced screening methodology. Using very well characterized inhibitors of Ab42 aggregation would be highly helpful, for example EGCG, bexarotene, specific chaperones (many additional examples are published in the literature with known mechanisms for targeting primary nucleation, secondary nucleation, elongation, or other aspects of Ab42 deposition). Seeing how inhibitors with well-characterized mechanisms change the Ab42 aggregation rate in their system would be very helpful for showing its utility.
- 5) How did the authors assess the cost comparison between the ThT system and the HaiDAP? The costs of the ThT assay

seems overstated, especially considering 384-well plates are commonplace.

6) The scale at which the Ab42 aggregates are imaged in this study is typically on the order of hundreds of microns. As such, limited structural insight into the types of Ab42 fibrils that are formed with and without inhibitors can be gained. The degree to which these Ab42 aggregates recapitulate those observed in AD at the structural level is not investigated, especially considering they contain Ab40 labeled with QDs. The TEM images are helpful, but do not resolve the above issues.

7) Details on how Ab42 was prepared are absent. All that seems listed in the methods is that human Ab42 was bought from rPeptide, and searching the provided catalog number did not readily show the kind of Ab42 used. Presumably it is recombinant, but with what counterion. How was it solubilized and monomerized? Was it size excluded or purified before use? Etc.

8) The rationale for the success rate difference of the plant extracts in PBS versus their system is lacking.

9) Further details on how B27 works to inhibit aggregation would be helpful.

Reviewer #2

(Remarks to the Author)

In the manuscript entitled „High-throughput screening technology for aggregation inhibitors of diseased cell-derived aggregative proteins (HaiDap) system”, Kuragano and colleagues present interesting data on the establishment of a higher throughput-compatible assay allowing to screen natural compounds for their ability to inhibit aggregation of Ab42 in an in vitro assay. This assay is facilitated by cell culture supernatant of hiPSC – derived neurons and can be evaluated in high throughput formats. Compared to previously presented assay systems that rely on PBS as buffer, this assay is more stringent in identifying crude plant extracts with such action. The assay system appears scalable and reproducible, as many experiments were performed with at least 3 replicate experiments.

- Although the Assay system presented identifies less active compounds, it is still somewhat surprising that more than 10% of the crude extracts showed activity and could be tested in a cellular assay system. A major point of concern is the cell-based assay system presented in the end of the manuscript. Important information (such as dose-dependency and / or toxicity readouts are not shown, even though they are mentioned in the text. Also, the HaiDap assay is supposed to “bridge” the less complex biochemical in vitro assays (such as the MSHTS) and more complex assays (such as the neuron-based system), and finally the effect in an animal model. Surprisingly, the assay fails to do so in the case of the only positive control substance (Rosmarinic Acid – RA) presented here: The authors show that RA is active in both assays, MSHTS, and HaiDap assay. It is stated that it shows a lower activity in HaiDap, but differences shown in Figure 5 are low, and as stated in the results section, HaiDap shows an approximately double EC50 compared to the PBS based system. Nevertheless, RA did not show any activity in the cell-based assay (Figure 7, Supplementary Figure 8 and 9). Also, authors noted / cited that RA is the only substance of the tested set that has repeatedly shown efficacy in animal models for amyloid deposition. In the eyes of this reviewer, this is worrisome, as the system was initially presented to be a better selector for efficacy in more complicated assays. If RA shows (albeit reduced) efficacy in HaiDap, why does it not show efficacy in the cell based model, but has so before in other studies in vivo?

Other specific comments / questions are listed below:

- Various experimental details are unknown / could not be easily evaluated by this reviewer. Please see below for some specific examples.
- One hiPSC line was used in this study. Therefore, saying that the HaiDap system is allowing a patient-specific screening is possibly overstated, this should be toned down.
- What is the rationale for including SHH protein in the differentiation protocol?
- There is a discrepancy between the materials & methods section and the legend of SupFigure 3 in regard to the medium changing / feeding procedure. That needs to be checked for consistency. Also, certain possibly important experimental details are missing, such as the coating procedure used for the cell culture vessels. Is the given cell seeding density at time of replating per well or cm² (see materials and methods)?
- The authors reported that aggregation in PBS (MSHTS) is faster and leads to a stronger signal than in NbM Alb(-). Figure 4 indicates that NbM Sup Alb(-) leads to an aggregation behaviour that is similar to the one obtained in PBS. This is interesting, as MSHTS is not as predictable as HaiDap according to authors. But what about the effect of the NbM Alb(-) on this selectivity? It is not clear to this reviewer that the “selectivity effect” of the HaiDap system is introduced by the culture medium, or by factors secreted by cells. Even though the aggregation in NbM Alb(-) is lower than in PBS, this selectivity effect should be checked in the cell-free system, but with the medium incubated on the culture vessel coating for the same time as with the cells. Adding B27 supplement to PBS, or just using Neurobasal medium would be also appropriate controls.
- It is not proven to this reviewer that the accelerated and increased aggregation is caused by Ab42 in the culture medium, as this is presented as an important factor. Medium cultured on cells and, for example, immunodepleted to remove Ab, would be an appropriate control. This is possibly even more relevant, as – to the understanding of this reviewer – the cell supernatant is used in the HaiDap assay with several orders of magnitude higher concentration of recombinant Ab42 added in the assay.
- Is the medium supernatant from the hiPSC-derived neurons collected from several medium changes? If so, is it pooled afterwards? This is potentially relevant, as the authors state in the discussion (page 8 – top part) that one benefit of the HaiDap system is to be more efficient than complicated hiPSC-based systems. If the medium is only collected once, one hiPSC culture would be needed for one harvest, largely reducing this effect.
- It is not easily understandable how the plant extracts were dissolved and applied in the assay system. Is it ensured that the dilution series is performed in the solvent, thus adding always the same amount of solvent into each dilution / the control wells? Also, in the cellular assay, DMSO is used as solvent, but it is not clear which concentrations are shown in the quantified figures, as well as what the replication of this data was (generating the error bars).

- Also, it is difficult to impossible to compare a purified substance (as RA) directly with a crude plant extract, such as done in the cell-based assay. Only one concentration is shown in the time-course of Figure 5, a dilution series would be needed to further specify the activity (see also below).
- Did the authors also evaluate other known compounds to inhibit aggregation, such as “beta sheet-breakers”? These would be an appropriate positive control for a revised manuscript.
- What do the cells in the cell-based assay system add to the assay? The selectivity for the compounds could come from the medium composition used, one possible benefit, using the cells for assessing toxicity of plant extracts, is briefly mentioned, but not shown. For the cell-based assays, it is therefore necessary to further elude on that, also how survival / toxicity was assessed.
- There are instances of figure duplication observed, as in Supplementary Figure 7: all plant extracts that do not show an effect in either system have always the same panels used for both MSHTS and HaiDap.
- Figure 4 is missing the legend for subpanel c.

Even though the system presented here is of interest to a scientific community, especially in the protein aggregation and Alzheimer’s Disease field, it is questionable if it is suitable for the very broad audience targeted by the journal in the current form. This in particular, as controls are missing to understand the origin of the higher selectivity of the HaiDap system and the disconnect of the positive control substance efficacy in two important assays presented here.

Reviewer #3

(Remarks to the Author)

Reviewer #4

(Remarks to the Author)

Version 1:

Reviewer comments:

Reviewer #1

(Remarks to the Author)

The authors have undertaken a solid revision with detailed rebuttals. Several questions remain, and I appreciate that the authors noted many of these limitations in the revised text. I remain concerned about not thoroughly addressing several of the limitations. In particular:

1. The concentrations of amyloidogenic proteins used (25 μ M for A β 42) are very high. The authors rationalized that “A β concentration was set at 25 μ M because at this concentration, the effect of A β aggregation on cells could be detected within 24 h.” I understand and appreciate the necessity to use higher concentrations of A β in vitro than in CSF, but cytotoxicity can usually be readily observed for cells treated with >1-2 μ M oligomers or fibrils after 15’ for single cell assays and easily in 20 h using bulk assays of cytotoxicity, and aggregation can monitored at the low micromolar concentration readily in many biophysical assays. The fact remains that the concentration of A β is quite high, which also limits potential ratios of A β to drugs (since the drugs will become cytotoxic).
2. I appreciate the authors “wanted to screen for substances that could potentially be ingested over a long period of time, so we screened 22 plant species in this study. These plant species were selected because they have a known history in the human diet and are relatively easy to obtain.” To me, the critical point is to prove the system reproducibly works, not to find a translational therapeutic from extracts. The fact that the authors are unable to show a molecule that works in the assays, therein adding confidence that the system works better and also comparing efficacies across the different methods, is concerning, especially considering very potent inhibitors/disaggregators exist (monoclonal antibodies, EGCG, beta sheet breakers as the other reviewer mentioned, many well characterized in vitro aggregation inhibitors that are commercially available like bexarotene).
3. The authors also write “Thus, it is expected that the HaiDap system will enable the evaluation of the inhibitory effects at each step of A β aggregation, such as seed formation and fibril elongation.” It is not clear how the authors would model molecular mechanisms like these microscopic steps with reproducibility from the total aggregated Ab data that are provided. Finding positive hits in their assays that have precise molecular mechanisms known in vitro would be helpful (some of which are known to have efficacy in vivo, eg monoclonal antibodies).
4. The authors clarify that the source of their Ab42 is synthetic, which adds to previous concerns that key aspect of this system likely do not recapitulate AD. This concern is amplified by point 1 above.
5. It remains unclear that the ab42 in the culture medium is what is accelerating/driving aggregation. Immunodepleting is a great suggestion from the other reviewer.
6. Minor point: Does G yesoense interfere with the QD signal, considering it starts muck lower than all the other samples in Fig 7D?

Considering the broad audience of the journal and that key aspects to validate the system remain unclear, I recommend this useful system be published in a more specialized journal.

Reviewer #2

(Remarks to the Author)

Thank you to the authors for providing detailed responses to the previous questions and comments. Especially for clarifying that the effects of in vivo of Rosmarinic Acid are rather secondary in nature and this can explain that RA is not identified as hit in this assay system presented here, even though it had shown in vivo efficacy. Most of the previous questions were addressed in sufficient detail, thank you for that. The following remaining points should be briefly addressed, though:

- It appears that the system or parts of the system were described in a patent application by some of the authors. If this patent is referenced in this revised version of the manuscript, it should be either listed in the references (if compatible with journal guidelines), or the data should be added as (supplementary) data to the manuscript. The patent application probably needs to be disclosed in the competing interests?
- Thank you for providing more detail on the differentiation of hiPSCs into neurons. Still, there is no reference given for including Shh protein in the maturation medium of the cells. Shh is a ventralizing patterning factor and will influence the outcome of the differentiation (regionalization) to a more ventral fate. If this differentiation procedure is based on a previously published procedure, it should be cited.
- If this reviewer understands the procedure correctly, no Glutamine was added in the Neurobasal medium (See also Table 1)? As the base medium formulation from the manufacturer is “(-) L-Glutamine”, indicating that this needs to be added, as also stated as recommended concentration of 0.5mM in the final medium. If Glutamine is not added, this should be briefly stated in the methods section to avoid confusion.

Reviewer #3

(Remarks to the Author)

Version 2:

Reviewer comments:

Reviewer #1

(Remarks to the Author)

The authors have added experiments to address the feedback of the reviewers. I remain concerned on the following points:

1. 25 μ M Ab42 is high as used in the MSHTS, and 5 μ M Ab42 is still high as used in the iPSC-based screening assay. More importantly, the use of synthetic peptide raises unanswered concerns. I don't think adding a limitation note is sufficient. Moreover, looking at the manufacturer's website, I cannot find purity details for item 4349-v (Ab42), and it does not appear this peptide was further purified (for example with size exclusion) by the authors beyond adding HFIP. Adding in the consideration that they are using Ab40QDs, the extent to which their aggregation reaction recapitulates key features of AD is unknown, which is a major claim of the paper. It would have been good if the authors had tested at least recombinant Ab42 of high purity for comparison (for example with EGCG or an extract as a positive control). Moreover, the use of only one male AD patient may further limit the potential relevance/utility/reproducibility in other labs of this study.
2. The experiments where 25 μ M monomer or oligomer are added to NbM Alb(-) and accelerate aggregation (Fig S6) support that Ab is relevant, but do not address the fundamental question of what in the iPSC supernatant is actually speeding up aggregation. The authors did not characterize their monomer or oligomer before testing it, so it is hard to be sure what species of Ab42 were actually added in that assay. This is further confounded by the point above. Gaining this clarity on what actually speeds up aggregation is a challenging point to address in the context of this paper, as the supernatant is necessarily complex, but it remains unclear what is secreted that dominates the observed effect (if it is even just Ab42 or some aggregated form of Ab42 - it likely has complex lipids/other metabolites co-aggregated, as the authors fairly suggest). Nonetheless, the lack of mechanistic insight remains an understandable limitation.
3. It would have been pragmatic to use the EGCG positive control to show dose-dependent inhibition in the iPSC-based screening assay.
4. An EC50 for EGCG in NbM Sup Alb(-) of 146.3 μ M is very high. IC50 values in the literature suggest it is in the low μ M range (1-10 μ M) against Ab42 in most in vitro assays. This discrepancy should be addressed. It also doesn't seem to agree with the potency reported in iPSC-based screening. Evidence rationalizing its efficacy in the various assays benchmarked to the literature is needed.
5. Minor: it is odd that S13b lacks the negative control without *Syzygium aromaticum* extract.

I commend the authors for their robust study and appreciate the challenges of working with iPSCs, both time and financial. Their concept and use of automation are sound and impressive, respectively. Despite these advances, the above concerns remain.

Version 3:

Reviewer comments:

Reviewer #1

(Remarks to the Author)

The author rebuttal is detailed and includes substantial new data (purification details of A β , HC controls, EGCG dose-response, and biochemical analyses). The work remains rigorous and addresses a major limitation in early-stage AD drug discovery. The contribution is of considerable technical novelty and creative. The paper is very well written.

The central thesis, that HiDap “will be useful for highly efficient, accurate, and low-cost screening of disease- and patient-specific drugs involved in protein aggregation,” remains undermined by the use of only one AD donor iPSC line. Although the authors acknowledge this limitation and include HC-iNeuron controls, the universality of the platform is unclear. The inclusion of the HC controls and their chemical compositions in Table 1 is very useful. I was surprised that aggregation was only modestly accelerated in Fig S7 in AD vs HC samples despite an approximately 8-fold increase in Ab40 and almost 4-fold increase in Ab42 from Table 1. The heterogeneity of the HC sample seems considerable, too, given the large error bars. The work runs the risk that the observed response is patient-specific or varies considerably from patient to patient, since they only tested one AD iPSC donor.

The main concern with synthetic Ab42, and recombinant for that matter, is that it likely contained significant impurities and, as I wrote in the last revision, it was not purified further by the authors in their lab using a technique like size-exclusion FPLC before kinetic assays were initiated. The recombinant 97% pure Ab42 used by the authors can contain significant impurities, including pre-formed Ab42 aggregates, without further purification, which does seem to be the case for both synthetic and recombinant peptides looking at Fig. R4 by SDS and silver staining. This, alongside using a very high concentration of Ab42, limits the conclusions that can be drawn as to mechanisms of action of hit molecules, but, the authors have not made over-stated claims on this point. Hopefully in future work, they can use lower concentrations of Ab42 that is of a purity sufficient for dedicated kinetic analyses.

Although the authors add HC vs AD supernatant comparisons and further A β characterizations, the supernatant remains complex and the mechanism of aggregation acceleration, whether caused some specific conformer or cell-modified version A β , lipids, metabolites, differences in pH, ionic strength, other uncharacterized biomolecules, or if it is due only to differences in the total secreted concentration of cellular Ab, remains uncertain. In their samples, ionic strength seems conserved and pH does seem to change only a small amount. I think there is enough evidence that the secreted Ab is important in light of their new experiments.

The work represents a positive step towards doing more relevant AD kinetic research. I hope they find their results universal in multiple AD donors, gain better understanding into how aggregation is accelerated and by what precise species, and resolve the reason for the differences in positivity/selectivity in the different assays (PBS, MSHTS, and HiDap).

Point-by-point responses to all reviewers' comments

Reviewer #1 (Remarks to the Author):

In the manuscript entitled "High-throughput screening technology for aggregation inhibitors of diseased cell-derived aggregative proteins (HaiDap) system," the authors have developed and refined a screening system for amyloid aggregation inhibitors using iPSCs derived from Alzheimer's disease patients. The goal of the paper is to study aggregation inhibitors in a more disease relevant context. This paper reports on the aggregation of A β 42, and other disease-associated amyloid proteins, in supernatant of the differentiated iPSC cell line, in addition to the kinetics of aggregation. The proposed methodology has optimized neural cells derived from an AD patient and reports the aggregation in presence of characterized culture media obtained from AD-derived neural cells. Cleverly, quantum dot labeled Ab40 with unlabeled Ab42 were used to define the shape, morphology and the texture of the aggregates at the scale of several hundreds of micrometers.

The scale to which the authors can screen compounds is impressive, and markedly higher throughput than most assays. I appreciate their efforts to do aggregation studies that are more biologically relevant to pathology, as this is important. However, a number of issues are noted below that in my opinion limit the probability that this system recapitulates key aspects of neurodegenerative disease.

We are honored that you have highly evaluated the HaiDap system. We carefully considered all comments, conducted additional experiments, and revised the manuscript. The point-by-point responses to all comments are listed below. We believe that this revision has significantly improved the manuscript and addressed your concerns.

1) The model system was established using cells derived from one AD patient (n=1). Beyond the limited n value, background information about the patient is missing, such as the genotype, if the patient was a familial or sporadic case of AD, age of onset, MMSE scores, etc.

We have added the following information about the iPS cell line derived from an AD patient to the "Cell culture" section of the Methods (Page 11, Lines 379-381); In this study we used one iPS cell line HPS0854 (RIKEN BRC, Tsukuba, Japan). Cell information; Cell line: HPS0854, Classification: iPSCs, Race: Japanese, Gender: Male, Age at sampling: 50s, Tissue: skin, Disease name: Alzheimer's disease.

In this study, we aimed to establish the basis of the HaiDap system by using the culture supernatant of iPSCs, and first used one iPS cell line. We believe that the usefulness of the HaiDap system developed in this study will be established by accumulating further screening knowledge using culture supernatants of various iPS cell lines. We added this information to the Discussion section as a limitation of this study and future prospects (Page 9, Lines 303-306).

2) Filtration also removes other proteins in solution than albumin (Fig. 2e), so it is possible that species beyond albumin are removed and likely are relevant to the aggregation kinetics of Ab42 since this is so dependent on solution conditions. While the secretion of species from the AD cells makes this system more relevant than standard immortalized cell lines, it is not known that this reaction mixture actually recapitulates key aspects of AD.

As the reviewer pointed out, filtration may remove molecules other than albumin, and it is not fully clear whether AD-iNeuron culture supernatants actually recapitulate important aspects of AD. On the other hand, we confirmed that approximately 80% of the A β ₄₂ secreted into the AD-iNeuron culture supernatant remains in the culture supernatant after filtration by optimizing the filtration technology (see US Patent App. 18/285,090, 2024). This study also demonstrated that screening with culture supernatant (HaiDap system) showed a higher correlation with subsequent iPSC-based assays than previous screening in PBS (MSHTS system). Based on the above, although it remains unclear to what extent the HaiDap system actually recapitulates key aspects of AD, we believe that it is more selective than previous screening in PBS. Information on the residual A β fraction after filtration has been added to the Results section (Page 4, Lines 130-132), while a reflection on whether this reaction mixture actually recapitulates key aspects of AD has been added to the Discussion section (Page 9, Lines 299-302).

3) The concentrations of amyloidogenic proteins used (25 μ M for A β 42) to test for aggregation in NbM sup Alb- medium are quite a bit higher than those used in typical in vitro aggregation studies (purified Ab42 typically aggregates with a half-time of 2 h at a concentration of 2 μ M). One would predict their system would show faster aggregation at such concentrations (eg a few μ M), considering the crowded nature of their media relative to a typical simple buffer system. It is unsurprising that aggregation begins immediately (Fig 7) considering the very high concentration of Ab42 and the crowded nature of the media. As a result, mechanistic insight into the microscopic mechanisms impacted by candidate compounds is limited in this system. Since the purpose of this system is to be more physiologically relevant, using at least 50-100x greater concentrations of Ab42 than are necessary to observe cytotoxicity is an issue.

As the reviewer pointed out, drug screening should be performed under physiological conditions as much as possible. On the other hand, the concentration of A β in cerebrospinal fluid is 0.186 nM, which is much lower than the 2 μ M used in general cell tests, as the reviewer pointed out. It is also unclear whether 2 μ M is optimal. We recently reported the kinetic verification of A β aggregation under viscous physiological conditions, kinetically demonstrating that it takes about 10 years for A β aggregates to grow to a diameter of 1 μ m when used at 0.186 nM (Kuragano et al., 2022; <https://doi.org/10.1016/j.colsurfb.2022.112449>). In fact, the size of amyloid plaques can be tens to hundreds of μ m in diameter, and this result is consistent with the knowledge that it takes several decades from A β aggregation to the onset of AD. Screening A β aggregation inhibitors over decades would take too long, so some form of acceleration is necessary. The simplest way to accelerate the aggregation reaction is to increase the reaction temperature or the concentration of molecules. However, since proteins are thermally denatured, we believe that the most appropriate way to accelerate the reaction is to increase protein concentration. In our previous paper on the development of an automated MSHTS system, we confirmed that 25 μ M A β aggregation saturated in approximately 16 h (Sasaki et al., 2019; <https://doi.org/10.1038/s41598-019-38958-0>). We also reported that in the presence of model neuronal PC12 cells, 20 μ M A β aggregated and deposited around the cells within 16 h, causing cell death (Kuragano et al., 2020; <https://doi.org/10.1038/s41598-020-66129-z>). As our objective was to establish a screening system that bridges in vitro screening to cell-based screening, A β concentration was set at 25 μ M because at this concentration, the effect of A β aggregation on cells could be detected within 24 h.

4) The rationale for using plant extracts for screening is not clear. The authors should consider elaborating on the plant extracts that were screened and provide additional background information on the significance of deploying specific plant extracts for this advanced screening methodology. Using very well characterized inhibitors of A β aggregation would be highly helpful, for example EGCG, bexarotene, specific chaperones (many additional examples are published in the literature with known mechanisms for targeting primary nucleation, secondary nucleation, elongation, or other aspects of A β deposition). Seeing how inhibitors with well-characterized mechanisms change the A β aggregation rate in their system would be very helpful for showing its utility.

We wanted to screen for substances that could potentially be ingested over a long period of time, so we screened 22 plant species in this study. These plant species were selected because they have a known history in the human diet and are relatively easy to obtain. Relevant statements have been added to the abstract (Page 2, Lines 38-39) and the Results section (Page 7, Lines 211-212).

Unfortunately, at this time we have not been able to identify any pure compounds that assayed positively in all three screening systems: PBS screening (MSHTS system), AD-iNeuron culture supernatant screening (HaiDap system), and the cell-based assay. The only compounds that were positive in all three screening systems were from the three plant extracts reported in this paper (*O. aristatus*, *S. aromaticum*, and *G. yesoense*). As the reviewer pointed out, we performed a cell-based assay using AD-iNeuron with two compounds, morin and quercetin, whose aggregation inhibition mechanisms are well known. However, as with RA, they were not as effective as the three plant extracts.

The results of the experiment are shown below:

The initial compound utilized was morin. Morin is a compound employed as a control reagent for Thioflavin T assays and is recognized for its capacity to impede seed formation from A β monomers. The other compound employed was quercetin, which is a compound that robustly inhibits A β fibril formation. In other words, it has been hypothesized that morin impedes early A β aggregation, while quercetin hinders later A β fibril formation to a greater extent than morin.

As shown below, the kinetics of A β aggregation inhibition were confirmed using these two compounds in a cell-based assay. It was determined that morin inhibits the early stage of A β aggregation (up to 6 h), while quercetin inhibits later A β fibril formation (after 19 h) to a greater extent than morin. However, the effect was not as dramatic as that of the three plant extracts.

Thus, it is expected that the HaiDap system will enable the evaluation of the inhibitory effects at each step of Aβ aggregation, such as seed formation and fibril elongation. However, no compounds used so far have shown dramatic effects, and at this stage, we were unable to explicitly mention these results in this paper.

5) How did the authors assess the cost comparison between the ThT system and the HaiDAP? The costs of the ThT assay seems overstated, especially considering 384-well plates are commonplace.

As the reviewer pointed out, it is also possible to perform the ThT assay using 384-well plates, so we clearly stated in the text that the cost calculations are for 96-well plates (Page 8, Line 271). We also mentioned the assay using 384-well plates in the legend of Supplementary Fig. 13.

6) The scale at which the Ab42 aggregates are imaged in this study is typically on the order of hundreds of microns. As such, limited structural insight into the types of Ab42 fibrils that are formed with and without inhibitors can be gained. The degree to which these Ab42 aggregates recapitulate those observed in AD at the structural level is not investigated, especially considering they contain Ab40 labeled with QDs. The TEM images are helpful, but do not resolve the above issues.

As the reviewer pointed out, we used QD-labeled Aβ₄₀ as a fluorescent probe, but the amount was 0.1% of Aβ₄₂. In our first paper in which this fluorescent probe was used, we confirmed that QDAβ₄₀, when added at a ratio of 0.1% to Aβ, was incorporated into Aβ₄₂ fibrils at a ratio of 0.1% and did not significantly affect the aggregation rate of Aβ₄₂ (Tokuraku et al., 2009; <https://doi.org/10.1371/journal.pone.0008492>). On the other hand, we also reported that QDAβ alone cannot form amyloid fibrils, probably due to steric hindrance by the QDs (Tokuraku and Ikezu, 2014; <https://doi.org/10.1016/B978-0-12-394431-3.00011-0>). Based on these prior results, we believe that the current imaging conditions using 0.1% QDAβ do not significantly affect Aβ₄₂ aggregation. Understandably, because this is an accelerated test in which Aβ concentrations are far higher than physiological conditions, concerns remain as to whether Aβ₄₂ aggregates recapitulate those observed in AD, as stated in response to comment 3 above. This is a dilemma when employing in vitro screening, and this concern can only be resolved by analyzing in detail the mechanism of inhibition of screened inhibitors in vivo, as in other screening methods.

7) Details on how Ab42 was prepared are absent. All that seems listed in the methods is that human Ab42 was bought from rPeptide, and searching the provided catalog number did not readily show the kind of Ab42 used. Presumably it is recombinant, but with what counterion. How was it solubilized and monomerized? Was it size excluded or purified before use? Etc.

The Aβ that was used can be easily searched for, and identified, when using the company name and product number (https://www.peptide.co.jp/en/catalog/f-cat?k_code=4349-v). To assist readers, we have added a brief explanation of the methods, as suggested by the reviewer (Page 10, Lines 325-328).

8) The rationale for the success rate difference of the plant extracts in PBS versus their system is lacking.

Following the reviewer's suggestion, we added a discussion of the differences in success rates between the MSHTS and HaiDap systems (Page 9, Lines 285-292).

9) Further details on how B27 works to inhibit aggregation would be helpful.

We performed additional experiments with B27 (Supplementary Fig. 1) and have added the results to the manuscript (Page 4, Lines 110-112).

Reviewer #2 (Remarks to the Author):

In the manuscript entitled „ High-throughput screening technology for aggregation inhibitors of diseased cell-derived aggregative proteins (HaiDap) system”, Kuragano and colleagues present interesting data on the establishment of a higher throughput-compatible assay allowing to screen natural compounds for their ability to inhibit aggregation of A β 42 in an in vitro assay. This assay is facilitated by cell culture supernatant of hiPSC – derived neurons and can be evaluated in high throughput formats. Compared to previously presented assay systems that rely on PBS as buffer, this assay is more stringent in identifying crude plant extracts with such action. The assay system appears scalable and reproducible, as many experiments were performed with at least 3 replicate experiments.

We thank you for your interest in our manuscript. We have carefully considered all of your comments, added some experiments to address your concerns, and revised the manuscript extensively. We believe that this revision has significantly improved the manuscript and addressed your concerns.

- Although the Assay system presented identifies less active compounds, it is still somewhat surprising that more than 10% of the crude extracts showed activity and could be tested in a cellular assay system. A major point of concern is the cell-based assay system presented in the end of the manuscript. Important information (such as dose-dependency and / or toxicity readouts are not shown, even though they are mentioned in the text. Also, the HaiDap assay is supposed to “bridge” the less complex biochemical in vitro assays (such as the MSHTS) and more complex assays (such as the neuron-based system), and finally the effect in an animal model.

As the reviewer pointed out, cell-based assays are more complicated than in vitro assays. In particular, cytotoxicity in the samples can significantly impact the results. To address the reviewer's concerns, we performed dose-dependent iPSC-based assays for the plant extracts that showed significant inhibitory activity (Supplementary Fig. 12) and added a corresponding explanation (Page 7-8, Lines 236-244). We believe that these experiments demonstrate that at least one plant extract, *S. aromaticum* extract, was effective in both the HaiDap system and cell-based assays.

Surprisingly, the assay fails to do so in the case of the only positive control substance (Rosmarinic Acid – RA) presented here: The authors show that RA is active in both assays, MSHTS, and HaiDap assay. It is stated that it shows a lower activity in HaiDap, but differences shown in Figure 5 are low, and as stated in the results section, HaiDap shows an approximately double EC50 compared to the PBS based system. Nevertheless, RA did not show any activity in the cell-based assay (Figure 7, Supplementary Figure 8 and 9). Also, authors noted / cited that RA is the only substance of the tested set that has repeatedly shown efficacy in animal models for amyloid deposition. In the eyes of this reviewer, this is worrisome, as the system was initially presented to be a better selector for efficacy in more complicated assays. If RA shows (albeit reduced) efficacy in HaiDap, why does it not show efficacy in the cell based model, but has so before in other studies in vivo?

As stated in response to comment 4 of reviewer #1, we have not been able to identify a pure compound that shows clear positive results in the MSHTS system, the HaiDap system, and the iPSC-based assay. RA, morin, and quercetin all showed a slight effect in iPSC-based assays, although the effects were not as great as those of the three plant extracts used in this study. In iPSC-based assays, factors such as inhibition of A β aggregation as well as deposition on the cell surface are involved, and it is believed that a mixture containing multiple active compounds, such as a plant extract, would be more advantageous than a single pure compound. The HaiDap system using QD imaging is also suitable for screening crude extracts containing various molecules, such as plant extracts (see the following paper for details; Sasaki et al., 2019; <https://doi.org/10.1038/s41598-019-38958-0>), and may be suitable for screening mixtures that effectively inhibit A β aggregation and deposition on neuronal surfaces.

As mentioned in the manuscript (Page 6, Lines 193-194), RA is a positive control in the MSHTS system using PBS, so we do not believe that it is a positive control that will be effective in all tests. In fact, the paper by Hase et al. (Ref. No. 29) cited in this paper reported that A β aggregation was suppressed by monoamines that increased in the brain following RA intake, and that RA does not directly suppress A β aggregation in the brain. To avoid any confusion to readers, we added some discussion about the experimental results using RA (Page 9, Lines 294-298).

Other specific comments / questions are listed below:

- Various experimental details are unknown / could not be easily evaluated by this reviewer. Please see below for some specific examples.

Following the reviewer's suggestions, we increased and fortified the descriptions of the experimental conditions in the Methods section (indicated in red text) and added an Excel file containing the source data for all experiments.

- One hiPSC line was used in this study. Therefore, saying that the HaiDap system is allowing a patient-specific screening is possibly overstated, this should be toned down.

We agree with your comment. A similar point was made by reviewer #1's comment (1), so kindly refer to that response.

- What is the rationale for including SHH protein in the differentiation protocol?

SHH was added to induce AD-iNeuron derived from NPCs. Please see the revised Method section for more details (Page 12, Lines 397-401).

- There is a discrepancy between the materials & methods section and the legend of SupFigure 3 in regard to the medium changing / feeding procedure. That needs to be checked for consistency. Also, certain possibly important experimental details are missing, such as the coating procedure used for the cell culture vessels. Is the given cell seeding density at time of replating per well or cm^2 (see materials and methods)?

The Methods section has been revised according to the reviewer's comments.

- The authors reported that aggregation in PBS (MSHTS) is faster and leads to a stronger signal than in NbM Alb(-). Figure 4 indicates that NbM Sup Alb(-) leads to an aggregation behaviour that is similar to the one obtained in PBS. This is interesting, as MSHTS is not as predictable as HaiDap according to authors. But what about the effect of the NbM Alb(-) on this selectivity? It is not clear to this reviewer that the "selectivity effect" of the HaiDap system is introduced by the culture medium, or by factors secreted by cells. Even though the aggregation in NbM Alb(-) is lower than in PBS, this selectivity effect should be checked in the cell-free system, but with the medium incubated on the culture vessel coating for the same time as with the cells. Adding B27 supplement to PBS, or just using Neurobasal medium would be also appropriate controls.

Following the reviewer's suggestions, we added the two experiments described next:

(1) Effects of 37 °C incubation in A β aggregation using NbM Alb(-)

To confirm whether the incubation without iPS cells affects A β aggregation, we observed A β aggregates in NbM Alb(-) with or without incubation, noting that the SD values (amount of A β) were not different between these conditions. We mentioned our new results in the revised manuscript (Page 5, Lines 166-171). In addition, we added images and a graph as Supplementary Fig. 5 with an accompanying figure legend in the revised Supplementary Information.

(2) Effects of B-27 solution on A β aggregation

As the reviewer suggested, we verified the effect of B-27 solution in PBS. We observed A β aggregates in various concentrations of B-27 solution in PBS, finding that 0.4-10% B-27 solution inhibited A β aggregates, indicating that A β aggregates might be inhibited in the general medium condition (i.e., 0.5% B-27 solution). We added these new results to the revised manuscript (Page 4, Lines 110-112). In addition, we added corresponding images and a graph as Supplementary Fig. 1 with an accompanying figure legend in the revised Supplementary Information.

- It is not proven to this reviewer that the accelerated and increased aggregation is caused by Ab42 in the culture medium, as this is presented as an important factor. Medium cultured on cells and, for example, immunodepleted to remove Ab, would be an appropriate control. This is possibly even more relevant, as – to the understanding of this reviewer – the cell supernatant is used in the HaiDap assay with several orders of magnitude higher concentration of recombinant Ab42 added in the assay.

We believe that various low molecular weight compounds, including A β ₄₂ derived from AD-iNeuron, affect A β aggregation. As shown in Table 1, in addition to A β ₄₂, lactate, glutamine, and A β ₄₀ are secreted. In order to clarify which molecules in the culture supernatant affect A β aggregation, it is necessary to quantify the concentrations of various molecules present in the culture supernatant, which is a future challenge. We have revised the statement that only A β secreted into the culture supernatant affects aggregation to avoid any misunderstanding by readers (Page 6, Line 172).

- Is the medium supernatant from the hiPSC-derived neurons collected from several medium changes? If so, is it pooled afterwards? This is potentially relevant, as the authors state in the discussion (page 8 – top part) that one benefit of the

HaiDap system is to be more efficient than complicated hiPSC-based systems. If the medium is only collected once, one hiPSC culture would be needed for one harvest, largely reducing this effect.

As suggested by the reviewer, detailed information regarding the collection of AD-iNeuron culture supernatants has been added to the Methods section (Page 12, Lines 389-406).

- It is not easily understandable how the plant extracts were dissolved and applied in the assay system. Is it ensured that the dilution series is performed in the solvent, thus adding always the same amount of solvent into each dilution / the control wells? Also, in the cellular assay, DMSO is used as solvent, but it is not clear which concentrations are shown in the quantified figures, as well as what the replication of this data was (generating the error bars).

As for general screening, samples were prepared in our experiments so that each dilution/control well always contained the same solvent composition. We added information about the error bars of the cell-based assays to the figure legends.

- Also, it is difficult to impossible to compare a purified substance (as RA) directly with a crude plant extract, such as done in the cell-based assay. Only one concentration is shown in the time-course of Figure 5, a dilution series would be needed to further specify the activity (see also below).

In cell-based assays, cytotoxicity can have a significant effect on the results. We performed dose-dependent experiments in the range where cytotoxicity was not an issue and have added the results to the main text (Page 7, Lines 238-244) and Supplementary Fig. 12.

- Did the authors also evaluate other known compounds to inhibit aggregation, such as “beta sheet-breakers”? These would be an appropriate positive control for a revised manuscript.

We confirmed that the chaperones CRT, Erp57, and PDI show A β aggregation inhibitory activity in PBS, although these are chaperones that function intracellularly. Since our target in this study was inhibition of A β aggregation outside cells, we decided that they would not be suitable as controls. As stated in response to Reviewer #1's comment No. 4, we also examined morin and quercetin, but were unable to find any clear positive results in all screenings to date.

- What do the cells in the cell-based assay system add to the assay? The selectivity for the compounds could come from the medium composition used, one possible benefit, using the cells for assessing toxicity of plant extracts, is briefly mentioned, but not shown. For the cell-based assays, it is therefore necessary to further elude on that, also how survival / toxicity was assessed.

As mentioned above, we added a new experiment on cytotoxicity (Supplementary Fig. 12).

- There are instances of figure duplication observed, as in Supplementary Figure 7: all plant extracts that do not show an effect in either system have always the same panels used for both MSHTS and HaiDap.

As the reviewer pointed out, we made a mistake during the presentation of these images. We apologize and have revised them accordingly.

- Figure 4 is missing the legend for subpanel c.

As the reviewer indicated, we had made a mistake in the Fig. 4 legend. We have now mentioned panel c in the revised manuscript.

Even though the system presented here is of interest to a scientific community, especially in the protein aggregation and Alzheimer's Disease field, it is questionable if it is suitable for the very broad audience targeted by the journal in the current form. This in particular, as controls are missing to understand the origin of the higher selectivity of the HaiDap system and the disconnect of the positive control substance efficacy in two important assays presented here.

As mentioned above, we have not yet found a pure compound that is clearly positive in all assays, although RA has shown a weak correlation. This is because various factors may be involved in the inhibition of aggregation and deposition of the aggregates in a complex system where cells exist. In this study, we found a plant extract that showed strong activity in both the HaiDap system and the cell-based assay. We believe that this concern will be alleviated by screening various iPS cell lines using the HaiDap system and clarifying the inhibitory mechanism of the obtained compounds in future studies. We acknowledge the limitations of the current method and stated them in the Discussion section of the revised manuscript (Page 9, Lines 299-309).

Reviewer #3 (Remarks to the Author):

We appreciate your efforts.

Reviewer #4 (Remarks to the Author):

We appreciate your efforts.

Reviewer #1 (Remarks to the Author):

The authors have undertaken a solid revision with detailed rebuttals. Several questions remain, and I appreciate that the authors noted many of these limitations in the revised text. I remain concerned about not thoroughly addressing several of the limitations. In particular:

We agree with the reviewer's comments that some limitations were not sufficiently addressed in the previous version of the manuscript. After carefully considering the reviewer's comments, we conducted additional experiments and revised the manuscript accordingly. We believe that the revised version fully addresses all of the reviewer's concerns. Thanks to the reviewers' insightful suggestions, we feel that the novelty and robustness of our paper have been greatly enhanced, and we are sincerely grateful for the constructive feedback.

1. The concentrations of amyloidogenic proteins used (25 μ M for A β 42) are very high. The authors rationalized that "A β concentration was set at 25 μ M because at this concentration, the effect of A β aggregation on cells could be detected within 24 h." I understand and appreciate the necessity to use higher concentrations of A β in vitro than in CSF, but cytotoxicity can usually be readily observed for cells treated with >1-2 μ M oligomers or fibrils after 15' for single cell assays and easily in 20 h using bulk assays of cytotoxicity, and aggregation can monitored at the low micromolar concentration readily in many biophysical assays. The fact remains that the concentration of A β is quite high, which also limits potential ratios of A β to drugs (since the drugs will become cytotoxic).

As pointed out by the reviewer, we used an A β concentration of 25 μ M in both the MSHTS and HaiDap systems because this can shorten the screening time and improve the sensitivity (Sasaki et al., 2019). On the other hand, in the cell-based assay, we used a concentration of **5 μ M A β** , which is in the same low micromolar range as recommended by the reviewer. In our previous manuscript, the A β concentrations used in the cell-based assays were only listed in the methods section (line 539). To make it clear to readers, we have now added this concentration information to the figure legends (lines 775-776) as well. Since both the 5 μ M A β concentration used in the cell-based assay and the 25 μ M concentration used in the HaiDap system are very high concentrations compared to physiological conditions, we have added a related discussion in the limitations section (lines 335-341). As we performed a dose-dependent iPSC-based assay

with *S. aromatica* extract (Supplementary Fig. 13), we believe that samples screened with the HaiDap system should be subjected to a careful second screening, including evaluation of cytotoxic activity in a subsequent iPSC-based assay. Therefore, the highly selective HaiDap system could be one of the excellent initial screening methods for rapidly finding useful substances that are effective *in vivo*.

2. I appreciate the authors “wanted to screen for substances that could potentially be ingested over a long period of time, so we screened 22 plant species in this study. These plant species were selected because they have a known history in the human diet and are relatively easy to obtain.” To me, the critical point is to prove the system reproducibly works, not to find a translational therapeutic from extracts. The fact that the authors are unable to show a molecule that works in the assays, therein adding confidence that the system works better and also comparing efficacies across the different methods, is concerning, especially considering very potent inhibitors/disaggregators exist (monoclonal antibodies, EGCG, beta sheet breakers as the other reviewer mentioned, many well characterized *in vitro* aggregation inhibitors that are commercially available like bexarotene).

Among the inhibitors recommended by the reviewer, we found EGCG to be a positive control that showed significant effects in all of the MSHTS system, HaiDap system, and cell-based assay. We have summarized the results in Supplementary Fig. 14 and added the relevant text at the end of the results (lines 258-268). We believe that this proves that the system works reproducibly.

3. The authors also write “Thus, it is expected that the HaiDap system will enable the evaluation of the inhibitory effects at each step of A β aggregation, such as seed formation and fibril elongation.” It is not clear how the authors would model molecular mechanisms like these microscopic steps with reproducibility from the total aggregated Ab data that are provided. Finding positive hits in their assays that have precise molecular mechanisms known *in vitro* would be helpful (some of which are known to have efficacy *in vivo*, eg monoclonal antibodies).

Regarding comments 3 and 5, we carefully considered the reviewer's comments and added a new experiment to respond to them. Specifically, we added A β monomers or oligomers at a concentration of 25 pM to NBM Alb(-) and observed A β aggregation (Supplementary Fig. 6).

The A β concentration of 25 pM is similar to the concentration of A β secreted from AD-iNeuron in NbM Sup and is also a physiological A β concentration. Interestingly, it was revealed that the addition of oligomeric A β accelerated the A β aggregation rate to the same extent as aggregation in NbM Sup Alb(-). These results indicate that A β secreted from the culture cell supernatant may form oligomers, which accelerate A β aggregation, and we believe that they represent part of the molecular mechanism of A β aggregation in NbM Sup Alb(-). We summarized the relevant results in Supplementary Fig. 6 and added explanatory text (lines 176-184).

4. The authors clarify that the source of their Ab42 is synthetic, which adds to previous concerns that key aspect of this system likely do not recapitulate AD. This concern is amplified by point 1 above.

Following reviewer comments, we added a limitation regarding the use of synthetic peptides (lines 331–335).

5. It remains unclear that the ab42 in the culture medium is what is accelerating/driving aggregation. Immunodepleting is a great suggestion from the other reviewer.

It is difficult to completely remove the target substance with the immunodepleting method, and non-specific adsorption may remove substances other than the target substance. Therefore, this time, A β was added in the form of monomers or oligomers at the concentration contained in the culture supernatant (NbM Sup Alb(-)) to the medium (NbM Alb(-)) and the effect on the aggregation rate and morphology was analyzed. For detailed results, please refer to the response to comment 3. These results showed that A β oligomers accelerate/drive the aggregation of A β ₄₂.

6. Minor point: Does G yesoense interfere with the QD signal, considering it starts muck lower than all the other samples in Fig 7D?

As the reviewer pointed out, I believe there is a possibility of interference, but even after subtracting this, it is clear that it inhibits aggregation.

Considering the broad audience of the journal and that key aspects to validate the system

remain unclear, I recommend this useful system be published in a more specialized journal.

We have carefully addressed your comments, which have allowed us to better clarify key aspects for validating our system. Protein-degenerative diseases, including AD, are increasing worldwide with the aging population. We believe this work will be of interest to the readers of Nature Communications, which targets a wide readership.

Reviewer #2 (Remarks to the Author):

Thank you to the authors for providing detailed responses to the previous questions and comments. Especially for clarifying that the effects of in vivo of Rosmarinic Acid are rather secondary in nature and this can explain that RA is not identified as hit in this assay system presented here, even though it had shown in vivo efficacy. Most of the previous questions were addressed in sufficient detail, thank you for that. The following remaining points should be briefly addressed, though:

Thank you very much for appreciating the last revision. We have added some experiments to address the further comments of Reviewer #1, and we believe that the paper is now more robust. We appreciate your helpful comments. We have addressed your comments as follows:

- It appears that the system or parts of the system were described in a patent application by some of the authors. If this patent is referenced in this revised version of the manuscript, it should be either listed in the references (if compatible with journal guidelines), or the data should be added as (supplementary) data to the manuscript. The patent application probably needs to be disclosed in the competing interests?

As suggested by the reviewers, we have added the relevant patent to the reference, and it has been disclosed in the competing interests.

- Thank you for providing more detail on the differentiation of hiPSCs into neurons. Still, there is no reference given for including Shh protein in the maturation medium of the cells. Shh is a ventralizing patterning factor and will influence the outcome of the differentiation (regionalization) to a more ventral fate. If this differentiation procedure is based on a previously published procedure, it should be cited.

Following reviewers' comment, we have added papers 51-53 to the Methods section (line 435).

- If this reviewer understands the procedure correctly, no Glutamine was added in the Neurobasal medium (See also Table 1)? As the base medium formulation from the manufacturer is “(-) L-Glutamine”, indicating that this needs to be added, as also stated as recommended concentration of 0.5mM in the final medium. If Glutamine is not added, this should be briefly stated in the methods section to avoid confusion.

Because glutamine-free Neurobasal medium was used in this AD-iNeuron induction step, we added a description of this in the Methods section (lines 436-437).

Reviewer #3 (Remarks to the Author):

Thank you for your review. Your help has significantly improved the manuscript and made it suitable for publication in Nature Communications.

Reviewer #1 (Remarks to the Author):

The authors have added experiments to address the feedback of the reviewers. I remain concerned on the following points:

To address the concerns raised by reviewer #1, we have conducted new experiments and attained additional verifications, including the use of recombinant A β , assessment using iPSC cells derived from a healthy subject, as well as an EGCG concentration-dependent analysis. We would like to express our sincere gratitude to reviewer #1 for their valuable advice.

1. 25 μ M Ab42 is high as used in the MSHTS, and 5 μ M Ab42 is still high as used in the iPSC-based screening assay. More importantly, the use of synthetic peptide raises unanswered concerns. I don't think adding a limitation note is sufficient. Moreover, looking at the manufacturer's website, I cannot find purity details for item 4349-v (Ab42), and it does not appear this peptide was further purified (for example with size exclusion) by the authors beyond adding HFIP. Adding in the consideration that they are using Ab40QDs, the extent to which their aggregation reaction recapitulates key features of AD is unknown, which is a major claim of the paper: It would have been good if the authors had tested at least recombinant Ab42 of high purity for comparison (for example with EGCG or an extract as a positive control). Moreover, the use of only one male AD patient may further limit the potential relevance/utility/reproducibility in other labs of this study.

Reviewer #1 asserts that the synthetic peptides used in our experiments “do not appear to have been further purified,” based on the absence of purity information on the manufacturer's website. We inferred that reviewer #1 assumed the cytotoxicity of A β was weak because low-purity A β ₄₂ peptide was used. To clarify this point, we contacted Peptide Institute, Inc. for details regarding the purity of the synthetic A β (4349-v) employed in our study. Although the purity of each lot is not disclosed on the company's website, we obtained detailed analytical data for the peptide shown in Fig. R1. According to this information, the peptide used in this study was purified using reverse-phase chromatography, and its purity was confirmed by HPLC to be 97.4%. We have added peptide purification method and purity information to the Methods section (lines 378-380).

Analytical Data

Record: 210283 ; Date: 2021-7-16

Code:	4349	Lot No:	710616
Compound:	Amyloid β -Protein (Human, 1-42) (Trifluoroacetate Form) Asp-Ala-Glu-Phe-Arg-His-Asp-Ser-Gly-Tyr-Glu-Val-His-His- Gln-Lys-Leu-Val-Phe-Phe-Ala-Glu-Asp-Val-Gly-Ser-Asn-Lys- Gly-Ala-Ile-Ile-Gly-Leu-Met-Val-Gly-Gly-Val-Val-Ile-Ala (M.W. 4514.04) C ₂₀₃ H ₃₁₁ N ₆₅ O ₆₀ S		
Appearance	: White amorphous powder		
Elemental Analysis	Found C, 47.24 ; H, 6.03 ; N, 14.00 %		
Amino Acid Analysis	Acid Hydrolysis: 6M HCl, 110°C, 22h. * concHCl:TFA:H ₂ O=2:1:1, 150°C, 4h.		
	Asp (4) 4.03	Ser (2) 1.78	Glu (4) 3.99
	Ala (4) 4.00	Val (6) 5.92*	Met (1) 0.95
	Leu (2) 2.00	Tyr (1) 0.98	Phe (3) 3.02
	Lys (2) 2.04	NH ₃ (2) 2.29	Arg (1) 1.00
Assay(AAA)	: 80%		
Mass Spectral Analysis	: Exhibits correct MW		
Purity(HPLC)*	: 97.4%		

* Purification method: Reverse-phase chromatography

Computer generated document
Valid without signature

Peptide Institute, Inc.
Quality Control Dept.
www.peptide.co.jp

Shimadzu CLASS-VF V6.14 SP2 HPLC14 Area % Report
F131 名:C:\CLASS-VF\Method\4000\0914349 Amyloid β -Protein (1-42).met
F- 名:C:\CLASS-VF\Data\210714\210714-002
分析日時:2021/07/14 11:55:36
印刷日時:2021/07/14 12:38:23

Sample : 4349 Amyloid β -Protein (Human, 1-42)
Sample Size : 0.5 μ L (0.52 mg/ 52 μ L-DMSO)
Column : XBridge Peptide BEH C18 300 \AA 3.5 μ m (4.6 mm I.D. \times 150 mm) #01273928316013
Eluent : 0.1% TFA
Gradient : Acetonitrile 20% to 60% [25 min.]
Flow Rate : 1.0 mL/min. ; Press. : 10.1 MPa ; Temp. : 60°C
Detection : CLK1 220 nm

Fig. R1 Analytical data of synthetic A β 42 (#4349-v) purchased from Peptide Institute, Inc.

To scientifically verify reviewer #1's concern that 5 μ M A β 42 is too high for cell-based assays due to its cytotoxicity, we purchased recombinant human A β 42 (#AG968, Sigma-Aldrich, MO, USA), which has been widely used in previous studies, as suggested by reviewer #1, and performed a cell-based assay. The purity of the purchased recombinant human A β was >97.0% (Fig. R2), comparable to the synthetic A β used in this study.

Sigma-Aldrich.

Certificate of Analysis

β Amyloid 1-42, a β , ultra-pure, HFIP, recombinant human

Cat.# AG968-1MG Pack Size: 1MG
 Lot # 4315258 Storage: -20°C

FOR RESEARCH USE ONLY
 NOT FOR USE IN DIAGNOSTIC PROCEDURES
 NOT FOR HUMAN OR ANIMAL CONSUMPTION

Counter Ion	HFIP
Peptide content	500 μ g per vial; no salt as HFIP desalts the sample.
Purity	>97.0%
Molecular Weight	4514.2 Da by Mass Spec Analysis (theoretical 4514.10)
Sequence	Asp-Ala-Glu-Phe-Arg-His-Asp-Ser-Gly-Tyr-Glu-Val-His-His-Gln-Lys-Leu-Val-Phe-Phe-Ala-Glu-Asp-Val-Gly-Ser-Asn-Lys-Gly-Ala-Ile-Ile-Gly-Leu-Met-Val-Gly-Gly-Val-Val-Ile-Ala
Presentation	Clear dry film. Resuspend in 1% NH ₄ OH at a concentration of 0.1 mg/mL-1 mg/mL. Sonicate for 30 seconds to 1 minute after it has gone into solution. To bring into your solution: After resuspension, add 5x or 10x buffer stock (PBS, TBS) and water to bring to 1x buffer.

The life science business of Merck KGaA, Darmstadt, Germany operates as MilliporeSigma in the US and Canada.

Submit your published journal article and earn credit toward future EMD Millipore purchases. Visit www.emdmillipore.com/publicationrewards to learn more!
 EMD Millipore Corporation, 28820 Single Oak Drive, Temecula, CA 92590, USA 1-800-437-7500
 FOR RESEARCH USE ONLY. Not for use in diagnostic procedures. Not for human or animal consumption. Purchase of this Product does not include any right to resell or transfer, either as a stand-alone product or as a component of another product. Any use of this Product for purposes other than research is strictly prohibited. Chemours, Upstate®, and all other trademarks, unless specifically identified above in the text as belonging to a third party, are owned by Merck KGaA, Darmstadt, Germany. Copyright ©2009-2020 Merck KGaA, Darmstadt, Germany. All rights reserved. 20774912; Version 1.0 Ver. 1.0 | 3MAY2025 | Cat.#AG968-1MG | AV

Fig. R2 Certificate of analysis of recombinant A β 42 (#AG968) purchased from Sigma-Aldrich.

The results indicate that, as reviewer #1 pointed out, recombinant A β was indeed more toxic than the synthetic A β used in our study (Fig. R3). Quantitative comparisons showed that the cytotoxicity of 0.5 μ M recombinant A β was comparable to that of 7.5 μ M synthetic A β , and that 2.0 μ M recombinant A β exhibited significantly higher cytotoxicity than these samples. These results support the previous comment of reviewer #1 that a 5 μ M A β concentration may be too high when using recombinant A β in iPSC-based assays. Finder et al. (2010, <https://www.sciencedirect.com/science/article/pii/S0022283609015216?via%3Dihub>) suggested that this difference in cytotoxicity may be due to impurities present during A β synthesis. However, according to a review by Varshavskaya et al. (2022, <https://www.mdpi.com/1422-0067/23/23/15036>), none of the reviewed studies, including Finder's work, examined the aggregation and structure of more than two A β species simultaneously, and a comprehensive comparison of these properties within a single study has yet to be conducted. Based on these findings, we considered it necessary to compare the biochemical properties of synthetic and recombinant A β with similar levels of purification, and therefore performed additional verification (Fig. R4).

Fig. R3 Evaluation of cytotoxicity of synthetic and recombinant Aβ peptide. For apoptosis assays of synthetic Aβ and recombinant Aβ; $1.8\text{--}2.2 \times 10^5$ cells/cm² of AD-iNeurons were incubated for 24 h in a NbM plus B27 medium containing synthetic or recombinant Aβ (0.5 μM, 2 μM, 7.5 μM). Apoptotic cells (in green) were subsequently detected using a fluorogenic substrate specific for Caspase-3/7 activity (Incucyte Caspase 3/7 green dye kit (#4440, Sartorius)). Treated plates were imaged at 20× magnification using phase and green channels, at nine images per well, and analyzed using Incucyte software.

a, Caspase 3/7 mediated apoptosis measured by fluorescence intensity. **b**, Merged images of brightfield and fluorescent field.

We performed SDS-PAGE to compare the biochemical properties of these Aβ species (Fig. R4a and R4b). In both samples, a major band corresponding to monomeric Aβ was observed (Fig. R4a). However, in the recombinant Aβ sample, additional bands were detected at molecular weights higher than that of the Aβ monomer, which appeared to originate from *E. coli* or Aβ oligomers (Fig. R4b). Notably, the recombinant Aβ band exhibited vertical streaks and appeared blurred—a common phenomenon observed during recombinant protein expression and purification, likely due to contaminating host DNA. When absorbance was measured using a NanoDrop 2000c spectrophotometer, recombinant Aβ showed absorption at 260 nm, indicative of nucleic acids (Fig. R5c). The A₂₆₀/A₂₈₀ ratio of recombinant Aβ was higher than that of synthetic Aβ (Fig. R5d), further suggesting that recombinant Aβ contains nucleic acids derived from *E. coli*. Although synthetic Aβ may also contain impurities that cannot be completely removed during purification, this study revealed that recombinant Aβ includes host-derived proteins and nucleic acids. These contaminants may influence cytotoxicity and aggregation under certain conditions, indicating that recombinant Aβ is not necessarily superior to synthetic Aβ for screening purposes. As summarized in the review by Varshavskaya et al., we believe it is essential to better understand differences in property based on peptide origin before use. Considering all these findings, the cytotoxicity of 5 μM synthetic Aβ used in our iPC-based assay is equivalent to that of 0.5 μM recombinant Aβ and is not as high as reviewer #1 feared. Furthermore, the significant differences observed between materials in the iPC-based assay using 5 μM synthetic Aβ do not undermine the validity of our experimental conditions. Therefore, in this paper we have decided to focus on

results using synthetic A β , with limitations regarding the peptides used for screening added in the Discussion section (lines 348-355).

Fig. R4 Biochemical validation of synthetic and recombinant A β peptides. **a** and **b**, SDS-PAGE of synthetic and recombinant A β ₄₂. To assess the purity of synthetic (S) and recombinant (R) A β ₄₂ peptides, samples prepared at a concentration of 100 μ M were separated by SDS-PAGE using a 16.5% acrylamide gel. After electrophoresis, bands were visualized in the gel by Coomassie Brilliant Blue (CBB) staining (**a**) and silver staining (**b**). Silver staining was performed using 2D-Silver Stain Reagent II (423413; Cosmo Bio Co., Ltd., Tokyo, Japan) according to the manufacturer's instructions. **c** and **d**, Absorbance measurements of synthetic and recombinant A β ₄₂ peptides. To evaluate the purity of synthetic and recombinant A β ₄₂, absorbance measurements were performed for both samples prepared at a concentration of 500 μ M (n = 3) using a NanoDrop 2000c Spectrophotometer (ND-2000c; Thermo Fisher Scientific Inc., Waltham, MA, USA) (**c**). The absorbance values at 260 nm and 280 nm were used to calculate the A260/A280 ratio, which was employed as an indicator of sample purity (**d**).

There are numerous examples of in vitro screening and cell-based assays using synthetic A β at concentrations of approximately 25 μ M and 5 μ M, respectively. Below, we present a selection of recent examples from reputable Springer Nature journals that employed synthetic A β peptides at similar concentrations. These examples demonstrate that our experimental conditions align with widely accepted practices in the field.

(1) Xiuhua Yin et al., Nature Communications, 14, 5718 (2023)

https://www.nature.com/articles/s41467-023-41489-y?utm_source=chatgpt.com

Synthetic A β peptides

In vitro aggregation inhibition assay : A β concentration 50 μ M

Cell based assay : A β concentration 50 μ M

(2) Qiang Luo et al., Nature Communications, 9, 1802 (2018)

https://www.nature.com/articles/s41467-018-04255-z?utm_source=chatgpt.com

Synthetic A β peptides

In vitro aggregation inhibition assay : A β concentration 20 μ M

Cell based assay : A β concentration 20 μ M

(3) K. Rajasekhar et al., Scientific Reports, 5, 8139 (2015)

https://www.nature.com/articles/srep08139?utm_source=chatgpt.com

Synthetic A β peptides

In vitro aggregation inhibition assay : A β concentration 20 μ M

(4) Si-Cong Bai et al., Communications Chemistry, 8, 196 (2025)

https://www.nature.com/articles/s42004-025-01587-y?utm_source=chatgpt.com

Synthetic A β peptides

In vitro aggregation inhibition assay : A β concentration 5 μ M

Cell based assay : A β concentration 5 μ M (10-fold dilution after incubation)

Table R1 Summary of the experimental conditions in the above papers.

Paper name	A β concentration		A β type
	In vitro aggregation inhibition assay	Cell based assay	
Xiuhua Yin et al. , 2023	50 μ M	50 μ M	Synthetic A β peptides
Qiang Luo et al. , 2018	20 μ M	20 μ M	Synthetic A β peptides
K. Rajasekhar et al. , 2015	20 μ M	—	Synthetic A β peptides
Si-Cong Bai et al. , 2025	5 μ M	5 μ M (10-fold dilution after incubation)	Synthetic A β peptides

Our simulations suggest that if A β aggregates at picomolar concentrations, it would take decades for the aggregates to reach an *in vivo* size of several micrometers, the size of senile plaques (Kuragano et al., 2022; <https://www.sciencedirect.com/science/article/pii/S0927776522001321>). It is well established that protein aggregation associated with protein-denaturing diseases occurs over very long timescales. Therefore, many researchers accelerate aggregation using high concentrations (Table R1), ultrasound, or other methods. Considering these reports and the experimental results described above, we believe that our experimental conditions fall within an acceptable range for the initial step of *in vitro* screening.

Reviewer #1 also expressed concern about the use of Ab40QD, although quantification of aggregation requires some kind of probe, such as ThT or an antibody. As mentioned in our initial response, we used QD-labeled A β 40 as a fluorescent probe; however, its concentration was only 0.1% relative to A β 42. This is an extremely

small amount, especially compared to the concentrations of ThT commonly used in amyloid aggregation assays, which are often equal to or greater than the peptide concentration. We also confirmed that adding QD-labeled A β ₄₀ at a 0.1% ratio did not significantly affect the aggregation kinetics of A β ₄₂, as demonstrated in our previous study (Tokuraku et al., 2009; <https://doi.org/10.1371/journal.pone.0008492>). Therefore, the use of trace amounts (0.1%) of QD-labeled A β ₄₀ is not considered to be particularly inferior to the use of ThT and anti-A β antibody, which has been reported to affect A β aggregation, and we believe that the use of trace amounts (0.1%) of QD-labeled A β ₄₀ does not impair the validity of the in vitro screening results. The reasons for not using QD-labeled A β ₄₂ and the advantages of using QDs for screening, such as the inner filter effect, the lack of fixation required for electron microscopy, and 3D analysis, are described in detail in the cited references (Ishigaki et al., 2013; <https://journals.plos.org/plosone/article?id=10.1371/journal.pone.0072992>, Sasaki et al. 2019, <https://www.nature.com/articles/s41598-019-38958-0>), including those mentioned above.

Following the comments of reviewer #1, we have also added a dose-dependent analysis using EGCG and experiments using different iPSC cell culture supernatants to the revised manuscript. We address these results in our response to reviewer #1's comment below.

2. The experiments where 25 pM monomer or oligomer are added to NbM Alb(-) and accelerate aggregation (Fig S6) support that Ab is relevant, but do not address the fundamental question of what in the iPSC supernatant is actually speeding up aggregation. The authors did not characterize their monomer or oligomer before testing it, so it is hard to be sure what species of Ab42 were actually added in that assay. This is further confounded by the point above. Gaining this clarity on what actually speeds up aggregation is a challenging point to address in the context of this paper, as the supernatant is necessarily complex, but it remains unclear what is secreted that dominates the observed effect (if it is even just Ab42 or some aggregated form of Ab42 - it likely has complex lipids/other metabolites co-aggregated, as the authors fairly suggest). Nonetheless, the lack of mechanistic insight remains an understandable limitation.

Following reviewer #1's suggestion, we have added results characterizing the monomers and oligomers used in the experiment (Supplementary Fig. 6d and 6e). Furthermore, we prepared healthy control (HC) iNeuron-derived NbM Sup Alb(-) and conducted a comparative study (Supplementary Fig. 7 and Table 1). The results revealed that the A β aggregation rate in HC iNeuron-derived NbM Alb(-), which contains lower levels of both A β ₄₂ and A β ₄₀ compared to the culture supernatant derived from AD iNeuron-derived NbM Alb(-) (Table 1), was slower than that observed in AD iNeuron-derived NbM Alb(-) (Supplementary Fig. 7). These findings suggest that iNeuron-derived A β and its oligomerization state influence A β aggregation. We have added these results to lines 192–206 in the Results section.

As reviewer #1 commented and as stated in the manuscript, it is possible that various molecules present in the culture supernatant affect A β aggregation. However, the addition of new experiments using HC iNeuron-derived NbM Alb(-), which has a relatively similar composition except for A β content, suggests that at least the amount and state of A β secreted from iNeurons derived from AD patients affect aggregation. We believe this strengthens our mechanistic insight.

3. It would have been pragmatic to use the EGCG positive control to show dose-dependent inhibition in the iPSC-based screening assay.

Following reviewer #1's instructions, we performed a dose-dependent iPSC-based screening assay using EGCG, and the results confirmed that the effect of EGCG was indeed dose-dependent (Supplementary Fig. 15d). These results have been added to line 279 of the Results section.

4. An EC50 for EGCG in NbM Sup Alb(-) of 146.3 μ M is very high. IC50 values in the literature suggest it is in the low μ M range (1-10 μ M) against Ab42 in most in vitro assays. This discrepancy should be addressed. It also doesn't seem to agree with the potency reported in iPSC-based screening. Evidence rationalizing its efficacy in the various assays benchmarked to the literature is needed.

As reviewer #1 pointed out, the magnitude of activity obtained by the HaiDap system and iPSC-based screening is not completely comparable, although samples in which activity was detected by the HaiDap system were also detected by iPSC-based screening. This is presumably due to direct interaction with the cell membrane and/or sample cytotoxicity, which is not affected by the HaiDap system. We have added this as a limitation of this study in the Discussion section (lines 335-340).

5. Minor: it is odd that S13b lacks the negative control without Syzygium aromaticum extract.

The negative control without extract in S13b (S14b in the new manuscript) is identical to the DMSO in the graph on the far right of Fig. 7d (A β intensity). Since it has already been presented, it is not displayed in S14b. However, for clarity, this information has been noted in the figure legend of S14b.

I commend the authors for their robust study and appreciate the challenges of working with iPSCs, both time and financial. Their concept and use of automation are sound and impressive, respectively. Despite these advances, the above concerns remain.

We thank you for recognizing our efforts. We believe that the HaiDap system has the potential to become a novel method that could dramatically accelerate the screening process for substances effective against

amyloidosis—a disease whose prevalence is increasing due to aging of the global population. To achieve this, further research involving a wide variety of iPS cell types and diverse peptides will be necessary; however, we believe this paper represents an important first step toward achieving that goal.

Reviewer #1 (Remarks to the Author):

The author rebuttal is detailed and includes substantial new data (purification details of A β , HC controls, EGCG dose-response, and biochemical analyses). The work remains rigorous and addresses a major limitation in early-stage AD drug discovery. The contribution is of considerable technical novelty and creative. The paper is very well written.

We appreciate your acceptance of our rebuttal, including the new data such as the purification details of A β , HC controls, EGCG dose–response experiments, and biochemical analyses. We also thank you for recognizing the technical novelty and creativity of this study.

The central thesis, that HiDap “will be useful for highly efficient, accurate, and low-cost screening of disease- and patient-specific drugs involved in protein aggregation,” remains undermined by the use of only one AD donor iPSC line. Although the authors acknowledge this limitation and include HC-iNeuron controls, the universality of the platform is unclear. The inclusion of the HC controls and their chemical compositions in Table 1 is very useful. I was surprised that aggregation was only modestly accelerated in Fig S7 in AD vs HC samples despite an approximately 8-fold increase in Ab40 and almost 4-fold increase in Ab42 from Table 1. The heterogeneity of the HC sample seems considerable, too, given the large error bars. The work runs the risk that the observed response is patient-specific or varies considerably from patient to patient, since they only tested one AD iPSC donor.

In response to this comment, we have incorporated the concerns raised by Reviewer #1 regarding the use of only one AD donor iPSC line into the Discussion section (lines 346–351). We have also revised the title and other parts of the manuscript to remove claims suggesting universal applicability.

The main concern with synthetic Ab42, and recombinant for that matter, is that it likely contained significant impurities and, as I wrote in the last revision, it was not purified further by the authors in their lab using a technique like size-exclusion FPLC before kinetic assays were initiated. The recombinant 97% pure Ab42 used by the authors can contain significant impurities, including pre-formed Ab42 aggregates, without further purification, which does seem to be the case for both synthetic and recombinant peptides looking at Fig. R4 by SDS and silver staining. This, alongside using a very high concentration of Ab42, limits the conclusions that can be drawn as to mechanisms of action of hit molecules, but, the authors have not made over-stated claims on this point. Hopefully in future work, they can use lower concentrations of Ab42 that is of a purity sufficient for dedicated kinetic analyses.

We have included a more detailed discussion of our concerns regarding synthetic and recombinant A β 42 in the Discussion section (lines 332–339), and we appreciate the reviewer’s acknowledgement of these limitations. We will address these issues carefully in future work.

Although the authors add HC vs AD supernatant comparisons and further A β characterizations, the supernatant remains complex and the mechanism of aggregation acceleration, whether caused some specific conformer or cell-modified version A β , lipids, metabolites, differences in pH, ionic strength, other uncharacterized biomolecules, or if it is due only to differences in the total secreted concentration of cellular Ab, remains uncertain. In their samples, ionic strength seems conserved and pH does seem to change only a small amount. I think there is enough evidence that the secreted Ab is important in light of their new experiments.

We have added a discussion of the potential mechanisms underlying aggregation promotion in the Discussion section (lines 351–354).

The work represents a positive step towards doing more relevant AD kinetic research. I hope they find their results universal in multiple AD donors, gain better understanding into how aggregation is accelerated and by what precise species, and resolve the reason for the differences in positivity/selectivity in the different assays (PBS, MSHTS, and HiDap).

We have also added a discussion (lines 354–358) addressing concerns regarding differences in positivity/selectivity among the different assays (PBS, MSHTS, and HiDap). As Reviewer #1 noted, this study represents a positive step toward more relevant research on AD aggregation dynamics. We hope to address the remaining concerns carefully in future studies and further enhance the platform’s usefulness for low-cost, high-throughput screening.